# Disentangling Latent Shifts of In-Context Learning Through Self-Training

## Abstract

In-context learning (ICL) has become essential in natural language processing, particularly with autoregressive large language models capable of learning from demonstrations provided within the prompt. However, ICL faces challenges with stability and long contexts, especially as the number of demonstrations grows, leading to poor generalization and inefficient inference. To address these issues, we introduce STICL (Self-Training ICL), an approach that disentangles the latent shifts of demonstrations from the latent shift of the query through self-training. STICL employs a teacher model to generate pseudo-labels and trains a student model using these labels, encoded in an adapter module. The student model exhibits weak-to-strong generalization, progressively refining its predictions over time. Our empirical results show that STICL improves generalization and stability, consistently outperforming traditional ICL methods and other disentangling strategies across both in-domain and out-of-domain data.

## 1 Introduction

In-context learning (ICL) (Brown et al., 2020) has emerged as a significant machine learning paradigm, particularly in natural language processing (NLP) applications that utilize large language models (LLMs). Unlike traditional supervised machine learning methods that rely on training over multiple epochs with large datasets, ICL leverages the ability of autoregressive LLMs to learn from context, with *demonstrations* and the *query* combined in a single prompt. This enables models to rapidly adjust to new tasks or varying input patterns without the need for additional fine-tuning. Moreover, ICL proves effective in low-resource setups by utilizing zero-shot and few-shot learning to perform tasks with minimal or no supervision (Dong et al., 2024a).

Despite its strengths, ICL faces several critical challenges. One of the key issues is **stability** – autoregressive LLMs based on the transformer architecture (Vaswani et al., 2017) can be highly sensitive to variations in the input context, such as the selection and ordering of demonstrations (Li et al., 2024; Lu et al., 2021; Dong et al., 2024a). This instability can result in poor generalization, making the models less reliable in real-world applications. Compounding this issue, ICL often involves long contexts because it requires incorporating multiple demonstrations alongside the query within a single input prompt. As more demonstrations are added, the input lengthens, and LLMs often struggle to handle extended contexts effectively. This problem can be traced to inherent primacy and recency biases, which lead models to overemphasize information positioned at the beginning or end of the context (Liu et al., 2024). Moreover, the inherent limitations of the context window size impose computational constraints, presenting a practical bottleneck (Dong et al., 2024b). Even with expanded context windows in newer models, the challenge of limited context persists. LLMs still struggle to fully utilize contexts when incorporating multiple demonstrations, often exceeding practical input lengths.

The aforementioned stability issues in ICL stem from the joint processing of demonstrations and the query. Since ICL can be viewed as introducing shifts in the model's internal representations – where knowledge from demonstrations is superimposed onto the latent features induced by the query – a promising solution is to **disentangle** these *latent shifts*, separating those induced by demonstrations from those of the query. By separating these shifts, ICL can process queries independently of demonstrations, reducing computational overhead and improving stability. Disentangling has been explored from various perspectives: Liu et al. (2023) and Zhang et al. (2024) have focused on

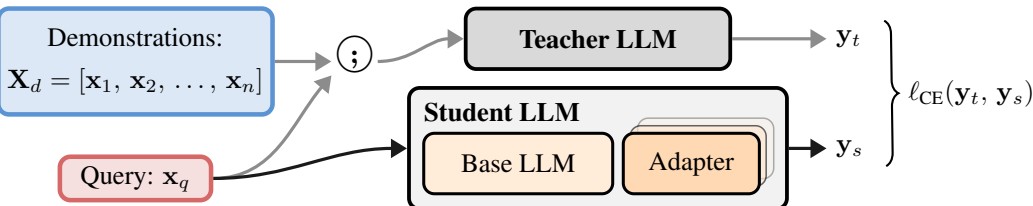

Figure 1: Illustration of STICL. The teacher processes a concatenation (denoted by $\odot$) of demonstrations $\mathbf{X}_d$, consisting of $n$ demonstrations $[\mathbf{x}_1, \mathbf{x}_2, \dots, \mathbf{x}_n]$, and the query $\mathbf{x}_q$. The student, using only the query, fine-tunes its adapter weights to produce outputs $\mathbf{y}_s$ aligned with the teacher's pseudo-labels $\mathbf{y}_t$ by minimizing the cross-entropy loss $\ell_{\text{CE}}$. After fine-tuning, the student can process only queries while still using the knowledge from demonstrations encoded in the adapter.

improving ICL's stability and scalability, while Dai et al. (2023) and Todd et al. (2024) leveraged disentangling to gain theoretical insights. Separating the latent shifts makes it possible to persistently store the context knowledge provided by demonstrations, eliminating the need to reprocess demonstrations with every query. This results in significantly shorter prompts, as only the queries remain, which can mitigate the problem of long context and improve the efficiency of inference. The latent shift induced by demonstrations can then be applied trivially, for example, by adding it to the latent features induced by the query. While disentangling the latent shifts of ICL has shown potential in improving ICL and advancing theoretical understanding, current methods rely on approximations, primarily by manipulating attention heads or hidden states. A more direct and principled approach to disentangling these shifts remains an open and compelling area for further investigation.

In this work, we propose to disentangle the latent shift of demonstrations from that of the query by explicitly focusing on the model's final outputs through the use of **self-training** (Amini et al., 2022). Self-training involves training a model using pseudo-labels generated by a previously learned model and has proven highly effective in leveraging unlabeled data for neural network training (Wei et al., 2021). We employ self-training in a simple teacher-student framework to encode the latent shift of demonstrations into a small set of additional parameters housed within an adapter module (Houlsby et al., 2019). Our method, STICL (**S**elf-**T**raining **ICL**), illustrated in Figure 1, employs a teacher LLM to generate pseudo-labels by processing both the demonstrations and the query without requiring extra labeled data. These pseudo-labels are then used to train a student LLM. The student model is trained to match the output provided by the teacher, taking only the query as input. By leveraging unlabeled data through self-training, the student can correct the pseudo-labels provided by the teacher, exhibiting weak-to-strong generalization (Lang et al., 2024). The method encodes the information from the demonstrations into the parameters and can seamlessly apply the latent shift just by activating the adapter module. Furthermore, due to the flexibility of adapters, a large set of demonstrations can be chunked into more manageable subsets, with each subset encoded in its own adapter module, and the modules can be easily merged. We evaluate STICL using autoregressive LLMs such as Llama 3 (8B) (Dubey et al., 2024) and Phi 3 (mini 4k) (Abdin et al., 2024) on the GLUE (Wang et al., 2018) and MMLU (Hendrycks et al., 2021) benchmarks, comparing it to pattern-based fine-tuning (Schick & Schütze, 2021) and few-shot ICL. On both in-domain (ID) and out-of-domain (OOD) data, STICL consistently outperforms these baselines and other disentanglement methods that leverage attention heads or hidden states, thus offering a reliable alternative without needing extra labeled data.

Our contribution is twofold: (1) We introduce STICL, a self-training ICL method that enhances efficiency and addresses stability and long-context challenges of ICL by disentangling the latent shifts between demonstrations and queries using one or several adapter modules; (2) We empirically demonstrate that STICL significantly improves both stability and generalization on ID and OOD, outperforming traditional ICL methods and other disentangling methods, while maintaining parameter efficiency. These findings suggest that even simple self-training setups, when properly designed, can offer substantial gains in ICL performance, paving the way for more efficient and scalable alternatives to current approaches.[1]

---

[1]The code is included in the supplementary material and will be made available upon publication.

## 2 METHOD

### 2.1 DISENTANGLING LATENT SHIFTS

Disentangling in-context knowledge from the query can aid in improving the efficiency and stability of ICL. Current approaches rely on manipulating the outputs of attention heads or hidden states. The motivation behind disentangling lies in previous research (Aizerman, 1964; Irie et al., 2022), demonstrating that linear layers optimized through gradient descent have a dual form of linear attention. To illustrate, consider a neural network's linear layer, where $\mathbf{W}_0, \Delta\mathbf{W} \in \mathbb{R}^{m \times n}$ denote the initial weight matrix and its subsequent updates by backpropagation, respectively. With $\mathbf{x} \in \mathbb{R}^m$ as the input representation, a linear transformation $\mathbf{f} : \mathbb{R}^m \to \mathbb{R}^n$ can be expressed as:

$$\mathbf{f}(\mathbf{x}) = (\mathbf{W}_0 + \Delta\mathbf{W})\mathbf{x}. \tag{1}$$

During backpropagation, $\Delta\mathbf{W}$ is computed by accumulating the outer products (denoted by $\otimes$) of $N$ training examples $\{\mathbf{x}_1, \mathbf{x}_2, \ldots, \mathbf{x}_N\}$, where $\mathbf{x}_i \in \mathbb{R}^m$, and the error signals $\{\mathbf{e}_1, \mathbf{e}_2, \ldots, \mathbf{e}_N\}$, where $\mathbf{e}_i \in \mathbb{R}^n$, obtained from the gradients of the loss function:

$$\Delta\mathbf{W} = \sum_{i=1}^{N} \mathbf{e}_i \otimes \mathbf{x}_i. \tag{2}$$

Irie et al. (2022) show that the update part of linear layers optimized by gradient descent can be expressed as unnormalized linear dot-product attention:

$$\mathbf{f}(\mathbf{x}) = (\mathbf{W}_0 + \Delta\mathbf{W})\mathbf{x} = \mathbf{W}_0\mathbf{x} + \sum_{i=1}^{N} (\mathbf{e}_i \otimes \mathbf{x}_i)\mathbf{x} = \mathbf{W}_0\mathbf{x} + \underbrace{\sum_{i=1}^{N} \mathbf{e}_i(\mathbf{x}_i^T\mathbf{x})}_{\text{linear attention}}. \tag{3}$$

In the context of the attention mechanism, this shows that the latent shift $\Delta\mathbf{W}\mathbf{x}$ corresponds directly to the application of linear attention, with error signals $\mathbf{e}_i$ as values, training examples $\mathbf{x}_i$ as keys, and the current input $\mathbf{x}$ as the attention query.

The concept of disentangling the latent shifts described in (3) can be extended to ICL, albeit only under the approximation of linear attention. Let $\mathbf{W}_V$, $\mathbf{W}_K$, and $\mathbf{W}_Q$ denote the weight matrices for values, keys, and queries, respectively. Let $\mathbf{x}_q^{(t)}$ represent the current query token's embedding at step $t$, and $\mathbf{q}^{(t)} = \mathbf{W}_Q\mathbf{x}_q^{(t)}$ is the corresponding attention query vector. The matrix $\mathbf{X}_q = [\mathbf{x}_q^{(1)}, \mathbf{x}_q^{(2)}, \ldots, \mathbf{x}_q^{(t-1)}]$ contains all previous query token representations up to $t - 1$, and $\mathbf{X}_d$ is the matrix of demonstration token representations. The concatenation $[\mathbf{X}_d; \mathbf{X}_q]$ along the sequence dimension is used to compute the attention output at step $t$, expressed as:

$$\mathbf{f}_{\text{AH}}(\mathbf{x}_q^{(t)}) = \mathbf{W}_V[\mathbf{X}_d; \mathbf{X}_q] \, \text{softmax}\left(\frac{(\mathbf{W}_K[\mathbf{X}_d; \mathbf{X}_q])^\top \mathbf{q}^{(t)}}{\sqrt{d}}\right), \tag{4}$$

where $d$ is the scaling factor (i.e., the dimensionality of the key vectors). By approximating the attention mechanism with linear attention, it becomes possible to disentangle the latent shift of the zero-shot output of an attention head induced by the query from the latent shift induced by the demonstrations (Dai et al., 2023):

$$
\begin{aligned}
\mathbf{f}_{\text{AH}}(\mathbf{x}_q^{(t)}) &\approx \mathbf{W}_V[\mathbf{X}_d; \mathbf{X}_q]\left(\mathbf{W}_K[\mathbf{X}_d; \mathbf{X}_q]\right)^\top \mathbf{q}^{(t)} \\
&= \underbrace{\mathbf{W}_V\mathbf{X}_q\left(\mathbf{W}_K\mathbf{X}_q\right)^\top}_{\mathbf{W}_{\text{ZS}}} \mathbf{q}^{(t)} + \underbrace{\mathbf{W}_V\mathbf{X}_d\left(\mathbf{W}_K\mathbf{X}_d\right)^\top}_{\Delta\mathbf{W}_{\text{ICL}}} \mathbf{q}^{(t)} \\
&= (\mathbf{W}_{\text{ZS}} + \Delta\mathbf{W}_{\text{ICL}})\,\mathbf{q}^{(t)}.
\end{aligned}
\tag{5}
$$

This approximation disentangles the latent shift induced by the demonstrations $\mathbf{X}_d$ from that induced by the query $\mathbf{x}_q^{(t)}$ (cf. Appendix A for detailed derivation of (5)). *The contribution from ICL is captured as a virtual weight update $\Delta\mathbf{W}_{ICL}$, corresponding to virtual gradients*, often referred to as "meta-gradients" in the literature. The zero-shot latent shift of the query, corresponding to $\mathbf{W}_{\text{ZS}}\mathbf{q}^{(t)}$, reflects the output without demonstrations, providing the initial state. Analogous to $\Delta\mathbf{W}\mathbf{x}$ in (3),

the latent shift $\Delta\mathbf{W}_{\text{ICL}}\mathbf{q}^{(t)}$ reflects the contribution of ICL. Finally, by substituting $\mathbf{h}_{\text{ZS}} = \mathbf{W}_{\text{ZS}}\mathbf{q}^{(t)}$ and $\Delta\mathbf{h}_{\text{ICL}} = \Delta\mathbf{W}_{\text{ICL}}\mathbf{q}^{(t)}$, we can rewrite the output of an attention head as:

$$\mathbf{f}_{\text{AH}}(\mathbf{x}_q^{(t)}) \approx \mathbf{h}_{\text{ZS}} + \Delta\mathbf{h}_{\text{ICL}}. \tag{6}$$

Although transformer-based LLMs use non-linear attention in practice, many approaches (Dai et al., 2023; Zhang et al., 2024; Todd et al., 2024) rely on the theoretical underpinnings of linear attention. These methods manipulate attention heads or hidden states to disentangle latent shifts despite the inherent non-linearity of the models. Furthermore, this simplification overlooks other crucial components of the transformer architecture, such as the feed-forward layers, activation functions, and residual connections. While approaches based on linear attention have proven effective, they leave room for further improvements in capturing and disentangling the full complexity of how transformers process data. In this work, we explore how virtual weight updates can be obtained more directly while preserving the key components of the transformer architecture.

## 2.2 SELF-TRAINING ICL

Building on the concept of disentangling latent shifts in transformer architectures, we introduce STICL (Self-Training ICL), an approach that offers a simple yet highly efficient way to internalize ICL knowledge into the parameters of a model. Rather than relying solely on manipulating attention heads, as is common in current methods, STICL aims to capture the full complexity of the transformer's components – considering the final output, which depends on all layers, including attention heads, feed-forward layers, and residual connections. By aligning more directly with the actual latent shifts induced by demonstrations, STICL ensures that the model uses the entirety of its architecture to first embed and later apply in-context knowledge.

At the core of STICL is a simple teacher-student framework: the teacher model, $\mathbf{f}_{\text{teacher}}$, processes both demonstrations and the query together to generate pseudo-labels without needing additional labeled data. The student model, $\mathbf{f}_{\text{student}}$, shares the same architecture as the teacher but includes adapter parameters. Unlike the teacher, the student processes only the query, using the adapter to internalize the knowledge from the demonstrations, as illustrated in Figure 1. Let $\mathbf{x}_q$ denote the query input and $\mathbf{X}_d$ the matrix of demonstration tokens, where each row corresponds to a single demonstration.[2] The empirical loss, defined using the cross-entropy loss $\ell_{\text{CE}}$, which operates on the teacher's output vector of probabilities for all tokens in the dictionary, is given by:

$$\sum_{\mathbf{x}_q \in \mathcal{D}_{\text{unlab}}} \ell_{\text{CE}} \left( \mathbf{f}_{\text{teacher}} \left( [\mathbf{X}_d^*; \mathbf{x}_q] \right), \mathbf{f}_{\text{student}} \left( \mathbf{x}_q \right) \right), \tag{7}$$

where $\mathcal{D}_{\text{unlab}}$ is an unlabeled dataset and $\mathbf{X}_d^*$ is a flattened version of $\mathbf{X}_d$. This approach is grounded in self-training (Amini et al., 2022), leveraging the teacher's pseudo-labels to fine-tune the student.

STICL fundamentally differs from existing approaches, which rely on manipulating attention heads or hidden states at query time. Instead, STICL progressively embeds the knowledge from demonstrations into the adapter parameters, denoted $\mathbf{W}_{\text{ICL}}$. The base LLM parameters, $\mathbf{W}_{\text{ZS}}$, capture the zero-shot component, while the total model parameters may be represented as $\mathbf{W}_{\text{ZS}} \oplus \mathbf{W}_{\text{ICL}}$, where $\oplus$ denotes the composition of base and adapter parameters.[3] This setup captures the latent shift introduced by the demonstrations through $\mathbf{W}_{\text{ICL}}$, extending the disentangling process outlined by (5) across the model's entire architecture. The teacher processes the full input sequence $[\mathbf{X}_d^*; \mathbf{x}_q]$, while the student processes only the query, applying $\mathbf{W}_{\text{ICL}}$ to integrate demonstration knowledge without explicitly processing the demonstrations. Analogously to (6), the latent shift induced by demonstrations can be recovered by decomposing outputs into zero-shot and ICL components. Let $\mathbf{h}_{\text{LLM}}(\mathbf{x}_q \mid \mathbf{W})$ represent the final latent states of an LLM with parameters $\mathbf{W}$ when processing the input $\mathbf{x}_q$. The following decomposition holds:

$$\mathbf{h}_{\text{LLM}}(\mathbf{x}_q \mid \mathbf{W}_{\text{ZS}} \oplus \mathbf{W}_{\text{ICL}}) = \mathbf{h}_{\text{LLM}}(\mathbf{x}_q \mid \mathbf{W}_{\text{ZS}}) + \Delta\mathbf{h}_{\text{ICL}}, \tag{8}$$

where $\Delta\mathbf{h}_{\text{ICL}}$ encapsulates the latent shift attributable to the demonstrations. STICL encodes the latent shift implicitly within the adapter parameters $\mathbf{W}_{\text{ICL}}$, which is central to our approach. However, if necessary, the latent shift can also be explicitly calculated owing to the decomposition in (8).

---

[2]The query $\mathbf{x}_q$ is a vector of token IDs, and $\mathbf{X}_d$ contains token IDs of demonstrations.

[3]Notably, the number of adapter parameters is significantly smaller compared to the base model parameters.

The stabilizing effect of STICL extends beyond just handling demonstrations. By iterating over multiple epochs, STICL leverages the same LLM instance for both the teacher and student roles, transitioning smoothly between them by activating or deactivating the adapter. Demonstrations can be shuffled across epochs to reduce sensitivity to their order, further stabilizing the ICL process. But the true power of STICL emerges from its parametric nature, which aligns with the optics of weak-to-strong generalization (Lang et al., 2024). The adapter parameters allow the model to internalize shifts and generalize effectively across both ID and OOD data, as demonstrated empirically in our experiments (cf. Section 3).

From the perspective of weak-to-strong generalization, the student model is not just expected to match the teacher – it is designed to outperform it. STICL facilitates this by leveraging *pseudo-label correction*, where incorrect labels are refined using high-confidence neighboring examples, and *coverage expansion*, enabling the model to generalize beyond regions initially covered by the teacher and even to near-OOD data (Section 3). STICL not only stabilizes ICL but also capitalizes on the parametric regime, where latent shifts can be efficiently encoded, enabling the model to establish implicit local-to-global consistency across the data distribution through extrapolation (Wei et al., 2021).

## 3 EXPERIMENTS

**Models.** We utilize a set of decoder-only autoregressive LLMs in our experiments. Specifically, we employ Hugging Face implementations (Wolf et al., 2020) of Llama 3 (8B) (Dubey et al., 2024) and Phi 3 (mini 4k) (Abdin et al., 2024) as our primary models, with additional comparison results for Llama 2 (7B) (Touvron et al., 2023). Detailed information about the models is provided in Table 12 in the Appendix.

**Evaluation.** We evaluate the models on the following benchmarks:

- **GLUE** (Wang et al., 2018): A standard benchmark for evaluating natural language understanding. We select the following datasets: four binary classification tasks for single sequences (COLA, SST, RTE), three binary classification tasks for sequence pairs (MRPC, QQP, QNLI), and one multi-class classification task for sequence pairs (MNLI). We follow the standard practice of evaluating models on the development sets. When evaluating generalization performance, we follow the standard practice and use Matthew's correlation for COLA, $F_1$ for MRPC and QQP, and accuracy for the remaining datasets;

- **MMLU** (Hendrycks et al., 2021): We evaluate the accuracy of multiple choice question answering on the MMLU benchmark, selecting two datasets with a sufficient number of instances for robust evaluation: "elementary math" (MATH), assessing basic mathematical reasoning skills, and "miscellaneous" (MISC), which covers diverse topics.

In our evaluation, we compute the first-token probability of the task verbalizers. We design the prompt template to guide the model toward generating the answer within the first token and limit the predictions to a subset of verbalizers (cf. Appendix F for details on prompt templates).

**Baselines and Methods.** We evaluate STICL by comparing it against three baselines and two ICL disentanglement methods:

- **Zero-Shot (0-shot)**: Predictions made without any demonstrations;

- **Standard ICL (n-shot)**: Utilizes $n$ demonstrations as context during inference;

- **Pattern-Based Fine-Tuning (PBFT)** (Schick & Schütze, 2021): Fine-tunes the model using patterns learned from data, framed as a language modeling task. In our experiments, we fine-tune an adapter module instead of the whole LLM;

- **In-Context Vectors (ICV)** (Liu et al., 2023): A forward pass is used on demonstration examples to create in-context vectors from the hidden states of the LLM;

- **Batch-ICL** (Zhang et al., 2024): Utilizes multiple separate one-shot forward computations and aggregates the resulting meta-gradients based on the attention head outputs.

Table 1: ID generalization scores for the 16-shot setup and $|\mathcal{D}_{\text{unlab}}| = 100$. The standard deviations of 10 runs are shown as subscripts. The highest scores and smallest standard deviations are highlighted in **bold**, while the second-best scores are underlined.

| Model | Method | GLUE | | | | | | | MMLU | |
|---|---|---|---|---|---|---|---|---|---|---|
| | | RTE | SST | QNLI | MNLI | COLA | MRPC | QQP | MATH | MISC |
| Llama 3 (8B) | 0-shot | 62.3 | 79.1 | 64.3 | 59.9 | 44.6 | 63.6 | 61.1 | 31.5 | 62.5 |
| | $n$-shot | $75.1_{6.5}$ | $93.5_{2.0}$ | $77.0_{5.5}$ | $68.0_{3.0}$ | $58.5_{4.0}$ | $74.0_{2.5}$ | $70.0_{3.0}$ | $43.5_{3.5}$ | $84.0_{4.0}$ |
| | PBFT | $73.2_{3.8}$ | $93.8_{1.5}$ | $77.8_{6.0}$ | $67.4_{3.5}$ | $56.5_{3.0}$ | $72.0_{2.0}$ | $68.0_{2.5}$ | $44.0_{3.8}$ | $83.5_{4.5}$ |
| | ICV | $72.9_{2.7}$ | $92.2_{1.8}$ | $74.5_{6.3}$ | $67.0_{4.2}$ | $57.3_{3.5}$ | $73.4_{2.3}$ | $69.1_{2.8}$ | $41.5_{4.3}$ | $67.0_{4.2}$ |
| | Batch-ICL | $77.8_{4.7}$ | $94.1_{2.2}$ | $78.0_{6.0}$ | $70.9_{3.5}$ | $59.8_{3.7}$ | $75.2_{2.2}$ | $\underline{72.5}_{2.7}$ | $36.2_{4.0}$ | $81.0_{2.5}$ |
| | STICL-F | $83.4_{0.3}$ | $95.1_{\mathbf{0.6}}$ | $\underline{80.3}_{\mathbf{1.4}}$ | $72.1_{2.5}$ | $\underline{63.7}_{\mathbf{1.5}}$ | $76.2_{1.8}$ | $71.9_{1.9}$ | $\underline{46.0}_{2.3}$ | $\underline{86.0}_{2.3}$ |
| | STICL-S | $\underline{86.0}_{\mathbf{0.6}}$ | $\mathbf{96.1}_{1.2}$ | $\mathbf{81.4}_{2.2}$ | $\underline{73.1}_{2.0}$ | $\mathbf{64.3}_{2.2}$ | $\mathbf{77.7}_{\mathbf{1.5}}$ | $\mathbf{73.1}_{1.8}$ | $\mathbf{49.5}_{\mathbf{2.0}}$ | $\mathbf{88.0}_{2.2}$ |
| | STICL-R | $\mathbf{86.5}_{3.0}$ | $\underline{95.5}_{0.8}$ | $79.0_{4.3}$ | $\mathbf{73.5}_{3.0}$ | $62.5_{2.8}$ | $\underline{76.5}_{1.9}$ | $72.0_{2.2}$ | $44.0_{2.7}$ | $85.5_{3.3}$ |
| Phi 3 (mini 4k) | 0-shot | 60.6 | 78.3 | 61.1 | 58.1 | 43.7 | 63.1 | 57.8 | 29.5 | 52.0 |
| | $n$-shot | $72.1_{5.2}$ | $90.6_{2.1}$ | $75.6_{3.2}$ | $65.3_{3.1}$ | $55.4_{4.1}$ | $71.1_{2.6}$ | $66.2_{3.7}$ | $37.5_{3.6}$ | $75.5_{4.1}$ |
| | PBFT | $70.6_{4.3}$ | $90.9_{1.9}$ | $73.6_{3.4}$ | $63.6_{3.6}$ | $53.6_{3.1}$ | $69.6_{2.3}$ | $64.6_{2.6}$ | $36.5_{4.1}$ | $73.5_{4.6}$ |
| | ICV | $71.5_{3.1}$ | $89.1_{2.1}$ | $74.3_{3.2}$ | $64.1_{4.1}$ | $54.1_{3.6}$ | $70.8_{2.4}$ | $65.4_{2.9}$ | $36.0_{4.6}$ | $74.0_{4.3}$ |
| | Batch-ICL | $75.3_{4.2}$ | $91.2_{2.6}$ | $76.6_{3.1}$ | $67.1_{3.6}$ | $56.1_{4.1}$ | $72.6_{2.6}$ | $67.3_{2.8}$ | $38.0_{3.9}$ | $76.0_{4.1}$ |
| | STICL-F | $\underline{80.4}_{1.2}$ | $92.1_{\mathbf{1.6}}$ | $\underline{78.2}_{\mathbf{1.3}}$ | $\underline{69.7}_{2.4}$ | $\underline{59.5}_{2.5}$ | $73.5_{2.1}$ | $\underline{68.6}_{2.2}$ | $\underline{40.5}_{3.2}$ | $\underline{77.5}_{3.6}$ |
| | STICL-S | $\mathbf{82.4}_{\mathbf{1.1}}$ | $\mathbf{93.2}_{1.6}$ | $79.2_{1.4}$ | $\mathbf{70.4}_{\mathbf{1.1}}$ | $\mathbf{60.7}_{2.3}$ | $\mathbf{74.1}_{\mathbf{1.4}}$ | $\mathbf{69.6}_{1.9}$ | $\mathbf{41.5}_{2.3}$ | $\mathbf{78.0}_{3.3}$ |
| | STICL-R | $79.0_{1.9}$ | $\underline{92.6}_{2.0}$ | $\mathbf{79.6}_{2.9}$ | $68.6_{3.9}$ | $58.6_{2.9}$ | $\underline{73.6}_{2.0}$ | $68.1_{2.3}$ | $39.5_{3.6}$ | $77.0_{3.7}$ |

In the experiments, we use $n \in \{4, 8, 16, 32\}$ instances for demonstrations and compare methods using a fixed number of demonstrations. Unless stated otherwise, we run each experiment 10 times with different seeds, which select different demonstrations in each run. In addition to the generalization scores, we report the standard deviation of the runs as an indicator of method stability. We evaluate performance on the GLUE development sets, while for the MMLU datasets, we sample 200 instances for evaluation.

**STICL variants.** We employ three variants of STICL, which differ in the variability of demonstrations they use, either in terms of selection or ordering:

- **STICL-Fixed (STICL-F)**: Uses a fixed set of demonstrations throughout training;
- **STICL-Shuffle (STICL-S)**: Shuffles the order of demonstrations at the start of each epoch;
- **STICL-Resample (STICL-R)**: Randomly resamples demonstrations before each epoch.[4]

We utilize LoRA (Low-Rank Adaptation) (Hu et al., 2022) for the adapter modules (for both PBFT and STICL), corresponding to 0.1–0.3% of the total parameter count, depending on the model (cf. Table 12 in the Appendix for adapter sizes per model). For each task, we generate pseudo-labels using the teacher model on unlabeled data. Specifically, we use 100 unlabeled instances ($\mathcal{D}_{\text{unlab}}$ in (7)) for both the GLUE and MMLU benchmarks. Additionally, for GLUE datasets, we experiment with 200 and 500 instances to assess the impact of the amount of unlabeled data on generalization and stability. We experiment only with 100 unlabeled instances for MMLU datasets due to their limited size. In all of the experiments, we fine-tune the adapter for 10 epochs. Further experimental details are provided in Appendix E.

## 3.1 GENERALIZATION AND STABILITY

We first evaluate the generalization and stability of STICL on ID data. Table 1 reports the 16-shot ID generalization scores along with standard deviations. Across all datasets and models, STICL-S consistently achieves the best generalization scores, outperforming standard ICL, PBFT, and the disentanglement methods ICV and Batch-ICL (cf. Table 5 in the Appendix for results with Llama 2). Compared to standard ICL, STICL-S *shows absolute improvements ranging from* 2.6% *to* 11.9% *for Llama 3 and* 2.5% *to* 10.3% *for Phi 3*, where the differences in scores are statically significant

---

[4]Although STICL-R uses the same number of demonstrations during inference as the other approaches, it requires access to a larger pool of labeled data since it draws new demonstrations in each epoch.

across all datasets.[5] Similar patterns hold for $n \in \{4, 8, 32\}$, where STICL-S also surpasses standard ICL (cf. Table 6 in the Appendix for other $n$-shot setups). Additionally, when a larger set $\mathcal{D}_{unlab}$ is used, there is a marginal improvement in scores, while stability improves even further (cf. Table 7 in the Appendix). Notably, the improvements in generalization with STICL-S, compared to standard ICL – the teacher model in STICL– provide strong evidence that the student model is exhibiting weak-to-strong generalization; we provide a more detailed analysis of this phenomenon in Section 4. While the STICL-F and STICL-R variants also show similar generalization scores as STICL-S, they generally exhibit higher variance compared to STICL-S, making STICL-S the preferred choice due to its higher stability with respect to demonstration selection – it improves upon standard $n$-shot ICL across all datasets and models. This is supported by the statistically significant differences in standard deviations on all datasets for Llama 3 and on all but QNLI for Phi 3.[6]

Having looked at stability with respect to demonstration selection, we now turn to a more focused evaluation of stability with respect to demonstration ordering. Table 2 reports the standard deviations across 50 runs, where the same set of demonstrations is used, but their order is shuffled for each run. Designed to adapt to shuffled demonstrations, STICL-S *shows the highest stability to demonstration ordering*, as evidenced by the smallest standard deviation. The stability improvements with STICL-S over standard ICL are statistically significant across all datasets.[6]

We next assess the capacity of STICL to perform OOD generalization by fine-tuning an adapter on one dataset and then applying the student model to a different dataset within the same task category, simulating a near-OOD scenario with pairs of closely related datasets. Table 3 shows the OOD generalization scores for such pairs of datasets in the GLUE benchmark. The results show that STICL-*S not only outperforms other methods in OOD generalization but also maintains higher stability when adapting to new domains* (cf. Table 8 in the Appendix for results with other models).

Table 2: Standard deviations of generalization scores across 50 runs with varied orderings of 16 demonstrations. The smallest deviations are in **bold**, and the second-smallest are underlined.

| Model | Method | GLUE | | | | | | | MMLU | |
| | | RTE | SST | QNLI | MNLI | COLA | MRPC | QQP | MATH | MISC |
|---|---|---|---|---|---|---|---|---|---|---|
| LLama 3 (8B) | $n$-shot | 4.81 | 1.62 | 4.19 | 2.22 | 3.04 | 1.81 | 2.03 | 2.52 | 2.87 |
| | PBFT | 2.71 | 1.14 | 4.53 | 2.69 | 2.27 | 1.57 | 1.82 | 2.70 | 3.22 |
| | ICV | 2.09 | 1.23 | 4.08 | 2.81 | 1.95 | 1.61 | 2.03 | 1.96 | 3.18 |
| | Batch ICL | 3.04 | 1.47 | 2.89 | 2.24 | 2.53 | 1.42 | 1.74 | 2.51 | 2.59 |
| | STICL-F | 1.32 | 0.72 | 1.53 | 1.83 | 1.76 | 1.54 | 1.38 | 1.89 | 2.07 |
| | STICL-S | **0.22** | **0.53** | **1.04** | **1.21** | **1.28** | **0.73** | **1.14** | **1.22** | **0.97** |
| | STICL-R | 2.04 | 1.34 | 2.47 | 2.05 | 1.85 | 1.48 | 1.64 | 2.03 | 2.51 |

Table 3: OOD generalization scores with 16 shots averaged over 10 runs, with standard deviations shown as subscripts. For each dataset pair, demonstrations are taken from the **left** dataset, and the model is tested on the **right** dataset. Columns represent results on the **right** datasets. The highest scores and lowest standard deviations are in **bold**, and the second-highest scores are underlined. Values in parentheses indicate differences from ID performance for the corresponding target dataset.

| Model | Method | QNLI $\rightarrow$ RTE | RTE $\rightarrow$ QNLI | QQP $\rightarrow$ MRPC | MRPC $\rightarrow$ QQP |
|---|---|---|---|---|---|
| Llama 3 (8B) | $n$-shot | $66.3_{2.4}$ (8.8) | $69.6_{1.3}$ (7.4) | $66.5_{1.9}$ (7.5) | $62.2_{2.3}$ (7.8) |
| | PBFT | $66.1_{1.5}$ (7.1) | $69.1_{1.6}$ (8.7) | $67.2_{1.8}$ (4.8) | $62.4_{1.2}$ (5.6) |
| | ICV | $65.7_{1.2}$ (7.2) | $68.7_{2.3}$ (5.8) | $67.5_{1.6}$ (5.9) | $63.0_{2.1}$ (6.1) |
| | Batch-ICL | $65.3_{1.4}$ (12.5) | $66.3_{2.5}$ (11.7) | $64.9_{2.3}$ (10.3) | $62.1_{2.1}$ (10.4) |
| | STICL-F | $\underline{67.5}_{1.1}$ (15.9) | $\underline{70.5}_{1.4}$ (9.8) | $68.5_{\mathbf{1.0}}$ (7.7) | $64.4_{1.5}$ (7.5) |
| | STICL-S | $\mathbf{69.0}_{\mathbf{0.5}}$ (17.0) | $\mathbf{71.3}_{\mathbf{0.7}}$ (10.1) | $\mathbf{69.0}_{2.2}$ (8.7) | $\underline{66.4}_{\mathbf{1.1}}$ (6.7) |
| | STICL-R | $67.1_{1.7}$ (19.4) | $70.0_{1.4}$ (9.0) | $68.0_{2.7}$ (8.5) | $\mathbf{68.3}_{2.0}$ (3.7) |

---

[5]We assess the statistical significance using a two-tailed Wilcoxon signed-rank test ($p < 0.05$), applying the Holm-Bonferroni method for family-wise error rate correction due to multiple comparisons.

[6] We evaluate significance using a two-tailed Levene's test ($p < 0.05$), applying the Holm-Bonferroni method for family-wise error rate correction.

## 3.2 ADAPTER ARITHMETIC

To overcome the limitations of context window sizes and efficiently handle extensive demonstration sets in ICL, we employ *adapter arithmetic* within STICL. STICL achieves this by fine-tuning separate adapters for each demonstration subset, with each adapter encoding the latent shift corresponding to its subset. These adapters are then merged by summing their parameters (Chitale et al., 2023), resulting in a single adapter that integrates knowledge from all subsets. Partitioning demonstrations into smaller subsets allows for better use of long contexts and effectively extending them without exceeding window limits or altering the base LLM architecture. Additionally, distributing the prompt across multiple adapters optimizes GPU utilization, fitting the entire prompt on a single GPU during inference and reducing memory constraints.

Table 4 shows the ID generalization scores of ICV, Batch-ICL, and STICL in fusing knowledge from multiple demonstration subsets, specifically using 2, 4, and 8 subsets of 16 demonstrations each. STICL-S consistently outperforms baseline methods, demonstrating its ability to fuse knowledge from different subsets. This success parallels broader trends in knowledge fusion within LLMs Wan et al. (2024). Moreover, this form of adapter arithmetic aligns with recent advances in task arithmetic, where merging task-specific parameters promotes generalization across multiple tasks (Ilharco et al., 2023; Ortiz-Jimenez et al., 2023). In our case, *this approach effectively improves generalization and stability when fusing demonstration subsets within the same task.*

Table 4: ID generalization scores of knowledge fusion for Llama 3. The scores are averaged over 10 runs with standard deviations shown as subscripts. The table compares the effectiveness of knowledge fusion from 2, 4, and 8 subsets of 16 demonstrations. The highest scores are in **bold**.

| Demonstrations | Method | GLUE | | | | | | | MMLU | |
|---|---|---|---|---|---|---|---|---|---|---|
| | | RTE | SST | QNLI | MNLI | COLA | MRPC | QQP | MATH | MISC |
| $2 \times 16$ | ICV | $75.2_{4.3}$ | $93.6_{1.9}$ | $77.6_{5.9}$ | $69.2_{3.7}$ | $58.3_{3.5}$ | $74.2_{2.4}$ | $70.6_{2.7}$ | $45.5_{3.7}$ | $72.5_{2.9}$ |
| | Batch-ICL | $80.2_{3.6}$ | $95.3_{1.8}$ | $80.2_{5.8}$ | $72.3_{3.0}$ | $61.2_{3.1}$ | $76.3_{2.0}$ | $72.6_{2.4}$ | $43.5_{2.9}$ | $83.0_{3.6}$ |
| | STICL-S | $\mathbf{87.1}_{1.6}$ | $\mathbf{96.4}_{1.3}$ | $\mathbf{81.5}_{5.0}$ | $\mathbf{75.5}_{2.5}$ | $\mathbf{68.4}_{1.8}$ | $\mathbf{78.5}_{1.4}$ | $\mathbf{74.1}_{1.6}$ | $\mathbf{51.5}_{1.6}$ | $\mathbf{89.5}_{2.0}$ |
| $4 \times 16$ | ICV | $78.3_{3.6}$ | $94.6_{1.8}$ | $79.3_{5.5}$ | $71.2_{3.1}$ | $60.3_{3.3}$ | $75.6_{2.2}$ | $72.3_{2.4}$ | $47.5_{3.5}$ | $76.5_{3.8}$ |
| | Batch-ICL | $84.4_{3.3}$ | $96.4_{1.5}$ | $82.4_{5.2}$ | $74.3_{2.5}$ | $64.2_{2.8}$ | $78.3_{1.6}$ | $74.3_{2.1}$ | $45.5_{2.6}$ | $84.5_{3.3}$ |
| | STICL-S | $\mathbf{88.4}_{2.3}$ | $\mathbf{97.5}_{0.7}$ | $\mathbf{83.6}_{4.4}$ | $\mathbf{77.3}_{2.2}$ | $\mathbf{71.4}_{1.5}$ | $\mathbf{79.6}_{0.7}$ | $\mathbf{75.2}_{1.3}$ | $\mathbf{53.5}_{1.4}$ | $\mathbf{91.0}_{1.7}$ |
| $8 \times 16$ | ICV | $81.3_{2.8}$ | $95.6_{1.5}$ | $81.8_{5.0}$ | $73.3_{2.7}$ | $61.3_{2.4}$ | $77.3_{1.7}$ | $73.8_{2.0}$ | $47.5_{2.9}$ | $78.0_{3.5}$ |
| | Batch-ICL | $85.6_{2.5}$ | $96.7_{1.1}$ | $83.8_{4.5}$ | $75.8_{2.1}$ | $65.3_{2.1}$ | $79.8_{1.3}$ | $75.8_{1.8}$ | $45.5_{2.0}$ | $84.0_{2.5}$ |
| | STICL-S | $\mathbf{92.8}_{0.8}$ | $\mathbf{98.1}_{0.2}$ | $\mathbf{87.9}_{2.5}$ | $\mathbf{81.3}_{0.9}$ | $\mathbf{74.1}_{0.6}$ | $\mathbf{82.8}_{0.4}$ | $\mathbf{78.9}_{0.5}$ | $\mathbf{57.0}_{0.5}$ | $\mathbf{93.0}_{0.7}$ |

## 4 ANALYSIS OF WEAK-TO-STRONG GENERALIZATION

Building on the observation that STICL consistently outperforms its teacher, standard ICL, we hypothesize that weak-to-strong generalization may be driving these improvements, where the model's ability to generalize strengthens progressively from weaker signals. To explore this further, we conduct an empirical analysis of STICL-S with Llama 3 on aggregated examples from all GLUE datasets, treating them as a single, unified dataset.

### 4.1 LOCAL CONSISTENCY

A crucial prerequisite for successful weak-to-strong generalization is the student's ability to maintain stable outputs under small perturbations of the input, i.e., robustness to input variations. A low Lipschitz constant serves as a key indicator of this stability, as it bounds the maximum change in the model output for any change in its input (Khromov & Singh, 2024). However, calculating the exact Lipschitz constant for LLMs is intractable. To approximate it, we leverage the relationship between the Lipschitz constant and the input-output Jacobian matrix of a neural network. Specifically, we compute the Frobenius norm of the Jacobian matrix as a tractable proxy, given its relationship to the spectral norm, which is a known lower bound for the Lipschitz constant (Dherin et al., 2022) (cf. Appendix B for theoretical details). Figure 2a presents the distribution of the approximated Lipschitz constants (normalized to $[0, 1]$) for STICL, PBFT, and ICL, providing a proxy for local consistency. STICL exhibits a notably lower Lipschitz constant than PBFT and ICL, underscoring its local consistency.

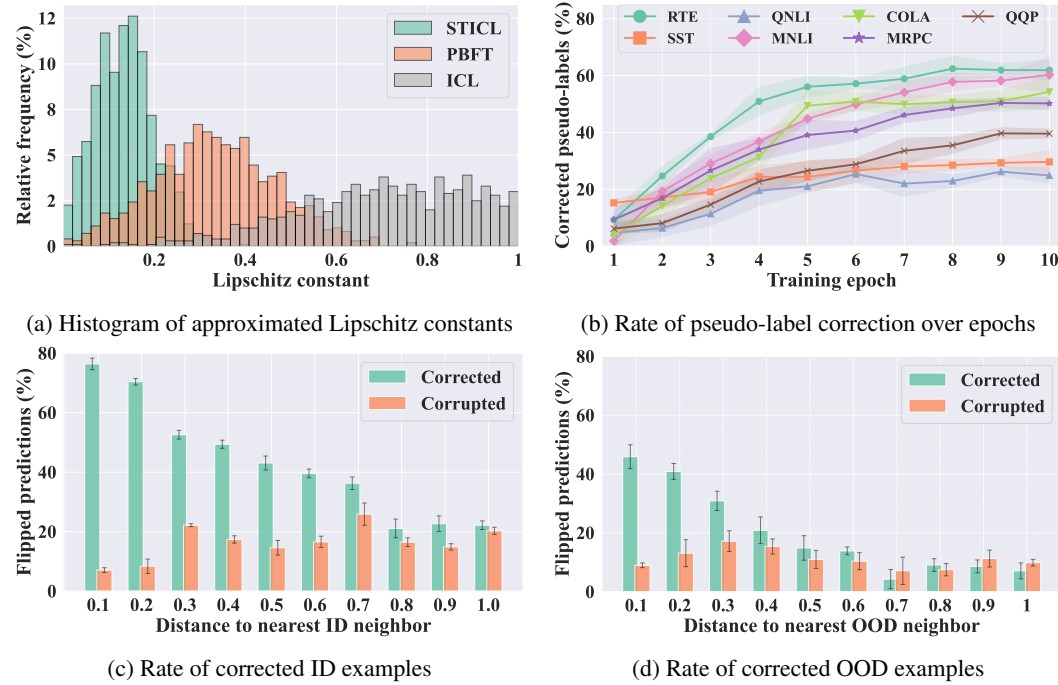

(a) Histogram of approximated Lipschitz constants

(b) Rate of pseudo-label correction over epochs

(c) Rate of corrected ID examples

(d) Rate of corrected OOD examples

Figure 2: Empirical analysis of STICL-S on the aggregated GLUE datasets for Llama 3: (a) Histogram of approximated Lipschitz constants across datasets, computed as the Frobeinus norm of the input-output Jacobian matrix; (b) Rate of pseudo-label correction over training epochs with examples from the unlabeled dataset used for self-training. Shaded areas indicate the standard deviation over 10 runs; (c) and (d) Corrected and corrupted prediction rates for (c) ID examples and (d) OOD examples, based on the Euclidean distance to the closest correctly pseudo-labeled neighbor (normalized to $[0, 1]$). There are 10 bins ranging from the interval of $[0, 0.1]$ to $[0.9, 1]$. Error bars denote the standard deviation over 10 runs.

## 4.2 PSEUDO-LABEL CORRECTION AND COVERAGE EXPANSION

*Pseudo-label correction*, where the student model revises the labels predicted by the teacher model, is a fundamental mechanism that drives weak-to-strong generalization (Lang et al., 2024). This process is closely tied to the model's ability to establish local consistency within the representation space, where accurate predictions in confident regions propagate corrections to neighboring, less certain areas, fostering local-to-global consistency throughout training. Figure 2b shows how the rate of corrected pseudo-labels evolves during training on GLUE datasets. As training progresses, the percentage of corrected pseudo-labels steadily increases, showcasing STICL's capacity to exhibit weak-to-strong generalization. Notably, the rate of pseudo-label correction plateaus faster for simpler datasets like SST and QNLI, which have lower linguistic variability.

The mechanism of pseudo-label correction ties into the phenomenon of *coverage expansion* – where the model generalizes beyond the regions covered by pseudo-labels Lang et al. (2024). We hypothesize that the core of STICL's ability to generalize effectively is anchored in coverage expansion, which enables local corrections to propagate globally, creating a ripple effect across the representation space. To understand this dynamic, we analyze which unseen evaluation points are corrected by clustering them based on their proximity to the nearest correctly pseudo-labeled neighbor in $\mathcal{D}_{\text{unlab}}$. This is quantified by computing the Euclidean distance between the model's representations at the final hidden states, with evaluation points categorized into ten bins based on their normalized distance from the correct neighbor, spanning the range $[0, 1]$. Figure 2c illustrates the rate of prediction flips within these bins, where a flip refers to either correcting an incorrect prediction or corrupting a correct one. The rate of corrected predictions shows a strong negative correlation with the distance to the nearest correctly labeled neighbor, as indicated by a Pearson correlation coefficient of $-0.968$, while corrupted predictions are more frequent in regions lacking nearby correct pseudo-labels.

Coverage expansion shows its effects even on OOD data. Figure 2d, the counterpart to Figure 2c, shows the rate of flipped predictions for OOD data. Although the impact is reduced, a similar correction pattern persists, with a Pearson correlation of $-0.916$. This consistency across domains highlights the model's ability to propagate accurate predictions not only within the training domain but also across OOD data.

## 5  RELATED WORK

**ICL theory.**    The understanding of ICL has shifted from a traditional task-learning framework to one focused on task identification. Wies et al. (2023) argue that ICL operates by recognizing latent tasks embedded within a model's pre-training, allowing for efficient performance on new tasks. Building on this, Hoogland et al. (2024) suggest that ICL in transformers progresses through distinct developmental stages, offering deeper insights into how models adapt to unfamiliar contexts. Li et al. (2023) further empirically show that ICL predictions become more resilient to input perturbations with longer prompts and that training on noisy data enhances stability. Despite these theoretical breakthroughs, ICL remains vulnerable to the selection and ordering of demonstrations (Li et al., 2024; Lu et al., 2021). Moreover, Kossen et al. (2024) highlight ICL's biases rooted in pre-training data, revealing that models do not always uniformly leverage in-context information.

**Disentaglement of latent shifts.**    Research into the inner workings of ICL has revealed how transformers process demonstrations to form task representations. Hendel et al. (2023) and Liu et al. (2023) show that transformers can compress demonstration examples into a task vector, which efficiently directs the model to generate context-appropriate outputs for queries. These task vectors are created during a forward pass, capturing the latent shift induced by the demonstrations. Building on this, Dai et al. (2023) explore using linear attention to compute virtual gradients, simulating the effect of gradient-based learning within the model. Similarly, Todd et al. (2024) use causal mediation analysis to highlight the role of specific attention heads in forming robust task representations in ICL, termed function vectors.

**Self-training and weak-to-strong generalization.**    Wei et al. (2021) provide a theoretical foundation for self-training, showing that under the assumption of coverage expansion, the minimizers of population objectives based on self-training and local consistency regularization achieve high accuracy. Lang et al. (2024) further develop the principle of pseudo-label correction, which occurs when the student model demonstrates strong local consistency. Several works have extended these ideas in the context of LLMs. For instance, Huang et al. (2023) demonstrate that LLMs can enhance their reasoning abilities through self-training without the need for labeled data by generating high-confidence, rationale-augmented answers, which are then used for fine-tuning, leading to improved performance across various tasks. In the same vein, Qu et al. (2024) propose recursive introspection for self-improvement, and Wang et al. (2024) introduce self-taught evaluators, showing how LLMs can autonomously refine and improve their outputs over time.

## 6  CONCLUSION

We tackled the challenges of stability and long-context handling that arise when processing multiple demonstrations in ICL within LLMs. To address these issues, we introduced STICL (Self-Training ICL), a method that disentangles the latent shifts induced by demonstrations from those of the query, leveraging a teacher-student framework. STICL encodes these latent shifts into an adapter module, enabling the student model to handle queries without requiring demonstrations in the input. Moreover, STICL allows efficient handling of large demonstration sets by chunking them into manageable subsets, each processed through separate adapter modules. This not only reduces the instability caused by demonstration selection and ordering but also alleviates the context window limitations inherent in transformer-based models. We demonstrated that STICL exhibits weak-to-strong generalization by refining pseudo-labels through progressive corrections, expanding from local consistency to a more comprehensive coverage across the representation space. Our empirical evaluation of STICL showed that it consistently outperforms traditional ICL methods, significantly improving generalization and stability across diverse datasets. These findings underscore the effectiveness of self-training as a promising strategy for improving ICL performance.

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

# A  DUAL FORM OF ICL

We offer a detailed derivation of (5), originally introduced by Dai et al. (2023), expanding on the key intermediate steps for clarity, which were not explicitly covered in the original work. The goal is to decompose the attention head output into separate components corresponding to the demonstrations and the query, thereby disentangling the latent shifts induced by ICL.

## A.1  STARTING POINT

We begin with the approximation of the attention head's output using linear attention:

$$\mathbf{f}_{\text{AH}}(\mathbf{x}_q^{(t)}) \approx \mathbf{W}_V[\mathbf{X}_d; \mathbf{X}_q]\left(\mathbf{W}_K[\mathbf{X}_d; \mathbf{X}_q]\right)^\top \mathbf{q}^{(t)}, \tag{9}$$

where:

- $\mathbf{W}_V \in \mathbb{R}^{d_h \times d_{\text{model}}}$ is the value weight matrix;

- $\mathbf{W}_K \in \mathbb{R}^{d_h \times d_{\text{model}}}$ is the key weight matrix;

- $\mathbf{X}_d \in \mathbb{R}^{d_{\text{model}} \times N_d}$ is the matrix of demonstration token representations;

- $\mathbf{X}_q \in \mathbb{R}^{d_{\text{model}} \times N_q}$ is the matrix of previous query token representations up to time $t-1$;

- $\mathbf{q}^{(t)} = \mathbf{W}_Q\mathbf{x}_q^{(t)} \in \mathbb{R}^{d_h}$ is the query vector at time $t$, with $\mathbf{W}Q \in \mathbb{R}^{d_h \times d_{\text{model}}}$ being the query weight matrix;

- $[\mathbf{X}_d; \mathbf{X}_q]$ is the concatenation of $\mathbf{X}_d$ and $\mathbf{X}_q$ along the sequence dimension.

## A.2  EXPANDING THE CONCATENATED MATRICES

We can expand the concatenated matrices as follows:

$$\mathbf{W}_V[\mathbf{X}_d; \mathbf{X}_q] = [\mathbf{W}_V\mathbf{X}_d; \mathbf{W}_V\mathbf{X}_q] = [\mathbf{V}_d; \mathbf{V}_q], \tag{10}$$
$$\mathbf{W}_K[\mathbf{X}_d; \mathbf{X}_q] = [\mathbf{W}_K\mathbf{X}_d; \mathbf{W}_K\mathbf{X}_q] = [\mathbf{K}_d; \mathbf{K}_q], \tag{11}$$

where:

- $\mathbf{V}_d = \mathbf{W}_V\mathbf{X}_d$ is the value matrix for the demonstrations;

- $\mathbf{V}_q = \mathbf{W}_V\mathbf{X}_q$ is the value matrix for the previous queries;

- $\mathbf{K}_d = \mathbf{W}_K\mathbf{X}_d$ is the key matrix for the demonstrations;

- $\mathbf{K}_q = \mathbf{W}_K\mathbf{X}_q$ is the key matrix for the previous queries.

The transpose of the concatenated key matrix is:

$$\left(\mathbf{W}_K[\mathbf{X}_d; \mathbf{X}_q]\right)^\top = \left[\mathbf{K}_d^\top; \mathbf{K}_q^\top\right]. \tag{12}$$

## A.3  PERFORMING THE MATRIX MULTIPLICATION

Substituting the expanded forms into Equation (9) using rules for block matrix multiplication, we have:

$$\mathbf{f}_{\text{AH}}(\mathbf{x}_q^{(t)}) \approx [\mathbf{V}_d; \mathbf{V}_q]\left[\mathbf{K}_d^\top; \mathbf{K}_q^\top\right]\mathbf{q}^{(t)} = \left(\mathbf{V}_d\mathbf{K}_d^\top + \mathbf{V}_q\mathbf{K}_q^\top\right)\mathbf{q}^{(t)}. \tag{13}$$

This separates the contributions from the demonstrations and the query sequences.

### A.4 DEFINING THE COMPONENTS

We define:

$$\mathbf{W}_{\text{ZS}} = \mathbf{V}_q \mathbf{K}_q^\top = \mathbf{W}_V \mathbf{X}_q \left( \mathbf{W}_K \mathbf{X} q \right)^\top, \tag{14}$$

$$\Delta \mathbf{W}_{\text{ICL}} = \mathbf{V}_d \mathbf{K}_d^\top = \mathbf{W}_V \mathbf{X}_d \left( \mathbf{W}_K \mathbf{X}_d \right)^\top. \tag{15}$$

Here:

- $\mathbf{W}_{\text{ZS}}$ represents the zero-shot component, capturing the model's behavior based on the query sequence alone;
- $\Delta \mathbf{W}_{\text{ICL}}$ represents the latent shift induced by the demonstrations, capturing the effect of in-context learning.

### A.5 FINAL EXPRESSION

Substituting (14) and (15) back into the expression, we obtain:

$$\mathbf{f}_{\text{AH}}(\mathbf{x}_q^{(t)}) \approx \left( \mathbf{W}_{\text{ZS}} + \Delta \mathbf{W}_{\text{ICL}} \right) \mathbf{q}^{(t)} = \mathbf{W}_{\text{ZS}} \mathbf{q}^{(t)} + \Delta \mathbf{W}_{\text{ICL}} \mathbf{q}^{(t)}. \tag{16}$$

### A.6 INTERPRETATION

The decomposition shows that the attention head output can be viewed as the sum of:

1. The **zero-shot component** ($\mathbf{W}_{\text{ZS}} \mathbf{q}^{(t)}$): the model's output when only the query sequence is considered, without any influence from the demonstrations;
2. The **latent shift due to ICL** ($\Delta \mathbf{W}_{\text{ICL}} \mathbf{q}^{(t)}$): the additional contribution from the demonstrations, representing the knowledge introduced via in-context learning.

This separation aligns with the theoretical motivation to disentangle the latent shifts induced by the demonstrations from those induced by the query, allowing for more efficient and stable processing of queries independently of demonstrations.

## B LIPSCHITZ CONTINUITY IN NEURAL NETWORKS

Lipschitz continuity is a fundamental concept in the analysis of neural networks as it provides a bound on how much the output of a function can change with respect to its input. Formally, a function $f : \mathbb{R}^n \to \mathbb{R}^m$ is said to be Lipschitz continuous with constant $L \geq 0$ if for any two inputs $\mathbf{x}, \mathbf{x}' \in \mathbb{R}^n$ the following inequality holds:

$$\|f(\mathbf{x}) - f(\mathbf{x}')\| \leq L \|\mathbf{x} - \mathbf{x}'\|.$$

This property ensures that the function $f$ behaves smoothly, meaning small changes in the input lead to small changes in the output, which is crucial for robustness in neural networks, particularly for predictive models (Khromov & Singh, 2024).

### B.1 RELATIONSHIP BETWEEN THE LIPSCHITZ CONSTANT AND THE JACOBIAN MATRIX

In neural networks, the Lipschitz constant can be bounded by the spectral norm of the Jacobian matrix, which quantifies the sensitivity of a function's output to changes in the input. The Jacobian matrix $\mathbf{J}_f(\mathbf{x}) \in \mathbb{R}^{m \times n}$ of a function $f$ is defined as the matrix of all partial derivatives:

$$[\mathbf{J}_f(\mathbf{x})]_{i,j} = \frac{\partial f_i(\mathbf{x})}{\partial x_j}.$$

The spectral norm of the Jacobian matrix, denoted $\|\mathbf{J}_f(\mathbf{x})\|_2$, provides an upper bound on the Lipschitz constant $L$ (Latorre et al., 2020):

$$\|\mathbf{J}_f(\mathbf{x})\|_2 \leq L, \forall \mathbf{x} \in \mathbb{R}^n.$$

The spectral norm represents the greatest possible rate of change in the function's output for any input variation. However, calculating the exact spectral norm can be computationally expensive, especially for deep neural networks, so the Frobenius norm is often used as an efficient alternative.

## B.2 Frobenius Norm as a Surrogate for the Lipschitz Constant

The Frobenius norm of the Jacobian matrix is often used as a surrogate for estimating the Lipschitz constant to avoid the computational complexity of calculating the spectral norm. The Frobenius norm, denoted $\|\mathbf{A}\|_F$, is easier to compute and relates to the spectral norm through the following inequality:

$$\|\mathbf{A}\|_2 \leq \|\mathbf{A}\|_F \leq \sqrt{r}\|\mathbf{A}\|_2,$$

where $r$ is the rank of the matrix $\mathbf{A}$. The Frobenius norm provides an upper bound on the spectral norm and thus serves as a useful proxy for estimating the Lipschitz constant. This approximation is particularly useful in large-scale models, such as LLMs, where direct computation of the spectral norm is infeasible.

## B.3 Empirical Evaluation of Lipschitz Continuity

In our experiments, we approximate the Lipschitz constant by computing the Frobenius norm of the input-output Jacobian matrix, where the embeddings are the inputs and the penultimate layer produces the outputs. As shown in Figure 2a, STICL demonstrates a significantly lower approximated Lipschitz constant compared to PBFT and ICL. This lower value suggests that STICL is more robust to input perturbations, which is a critical property for correcting pseudo-labels.

## C  Limitations

**Computational cost.**    STICL introduces additional computational overhead due to the fine-tuning of adapters during the self-training process. While this fine-tuning is more lightweight compared to full model fine-tuning, it remains more expensive than standard in-context learning (ICL), which avoids weight updates entirely. However, STICL offsets some of this cost by removing demonstrations from the input during inference. For instance, with Llama 3 (8B) processing 16 demonstrations from GLUE datasets, inference takes approximately 120 times longer than a 0-shot setup (processing only the query). This increased cost scales quadratically with the number of tokens, highlighting the self-attention mechanism as the primary bottleneck when handling 16 demonstrations. Based on our measurements, self-training with 100 unlabeled instances and 16 demonstrations using a single adapter corresponds to the computational cost of approximately 2100 inferences in a 16-shot setup. This implies that after about 2100 inferences, the time spent on fine-tuning is effectively balanced by the reduction in per-inference computational cost.

**Applicability.**    STICL may be less suitable for scenarios with extremely limited resources, as it relies on access to a supply of unlabeled data. In our experiments with $\{4, 8, 16, 32\}$ demonstrations, we typically used 100 unlabeled instances, which proved sufficient to achieve strong performance. While unlabeled data is generally easier to acquire than labeled data, there may be scenarios where obtaining even a modest amount of unlabeled data is challenging, potentially limiting the applicability of STICL.

**Large demonstration sets.**    Although STICL efficiently encodes demonstrations into adapters to overcome context length limitations, the method has not been extensively tested with very large demonstration sets. From our findings, as the total number of demonstrations increases, using multiple adapters with manageable demonstration sizes tends to be more effective. For instance, we successfully employed 8 adapters with 16 demonstrations each (totaling 128 demonstrations). While this approach theoretically allows for an indefinite increase in the number of demonstrations, its effectiveness with significantly larger sets remains unexplored. Moreover, using additional adapters increases computational costs, introducing a tradeoff between scalability and efficiency.

## D  Additional Results

### D.1  Supplementary Tables

Here, we present additional results that supplement those in the main paper.

Table 5: ID generalization scores for the 16-shot scenario and $|\mathcal{D}_{\text{unlab}}| = 100$ for LLama 2 (7B). The standard deviations of 10 runs are shown as subscripts.

| Model | Method | GLUE | | | | | | | MMLU | |
|---|---|---|---|---|---|---|---|---|---|---|
| | | RTE | SST | QNLI | MNLI | COLA | MRPC | QQP | MATH | MISC |
| Llama 2 (7B) | 0-shot | 57.8 | 75.4 | 59.3 | 55.7 | 40.7 | 59.4 | 58.7 | 29.0 | 59.0 |
| | $n$-shot | $69.2_{4.3}$ | $89.8_{2.1}$ | $74.2_{5.9}$ | $63.3_{2.8}$ | $54.3_{3.5}$ | $66.9_{2.4}$ | $64.7_{1.5}$ | $37.5_{4.8}$ | $80.0_{5.3}$ |
| | PBFT | $69.0_{2.7}$ | $89.7_{0.4}$ | $73.3_{5.0}$ | $64.4_{4.7}$ | $51.2_{2.9}$ | $67.9_{2.0}$ | $64.6_{1.6}$ | $40.0_{3.2}$ | $79.5_{2.1}$ |
| | ICV | $68.0_{4.6}$ | $87.8_{2.6}$ | $71.2_{6.7}$ | $60.9_{4.0}$ | $53.1_{2.4}$ | $68.8_{1.7}$ | $65.0_{1.9}$ | $39.5_{2.7}$ | $62.5_{0.6}$ |
| | Batch-ICL | $75.2_{0.8}$ | $91.2_{1.9}$ | $74.0_{0.8}$ | $66.5_{3.3}$ | $55.9_{2.1}$ | $70.3_{0.8}$ | $69.1_{1.8}$ | $34.5_{2.3}$ | $77.0_{4.1}$ |
| | STICL-F | $77.2_{0.7}$ | $90.2_{0.7}$ | $76.8_{4.2}$ | $66.5_{2.4}$ | $60.1_{1.2}$ | $71.6_{0.2}$ | $68.8_{0.8}$ | $43.0_{1.6}$ | $82.5_{2.5}$ |
| | STICL-S | $81.9_{2.5}$ | $92.1_{0.3}$ | $77.3_{0.9}$ | $70.4_{1.8}$ | $62.8_{3.4}$ | $72.3_{2.6}$ | $68.2_{0.5}$ | $46.5_{1.5}$ | $82.5_{1.7}$ |
| | STICL-R | $81.1_{1.9}$ | $93.6_{2.0}$ | $74.7_{3.6}$ | $69.6_{2.9}$ | $57.9_{2.9}$ | $73.1_{2.0}$ | $66.8_{2.3}$ | $41.5_{2.6}$ | $82.0_{3.7}$ |

Table 6: ID generalization scores for $n$-shot scenarios ($n = 4, 8, 32$, with $\mathcal{D}_{\text{unlab}} = 100$) for Llama 3 (8B). The standard deviations of 10 runs are shown as subscripts.

| Model | $n$ | Method | GLUE | | | | | | | MMLU | |
|---|---|---|---|---|---|---|---|---|---|---|---|
| | | | RTE | SST | QNLI | MNLI | COLA | MRPC | QQP | MATH | MISC |
| Llama 3 (8B) | 4 | $n$-shot | $71.3_{5.4}$ | $84.5_{4.4}$ | $70.1_{2.9}$ | $62.4_{2.7}$ | $54.6_{3.5}$ | $69.2_{4.1}$ | $62.0_{2.3}$ | $37.0_{3.9}$ | $76.5_{2.5}$ |
| | | STICL-S | $80.3_{1.5}$ | $90.9_{0.9}$ | $76.3_{1.4}$ | $70.1_{1.8}$ | $61.4_{2.0}$ | $72.9_{1.5}$ | $70.3_{1.2}$ | $43.0_{1.3}$ | $77.5_{1.8}$ |
| | 8 | $n$-shot | $72.7_{2.1}$ | $89.4_{2.6}$ | $73.5_{2.5}$ | $64.7_{3.1}$ | $55.8_{2.8}$ | $71.2_{2.4}$ | $64.3_{2.9}$ | $37.0_{1.3}$ | $77.5_{2.1}$ |
| | | STICL-S | $82.1_{1.1}$ | $93.2_{1.0}$ | $78.3_{1.3}$ | $72.2_{1.6}$ | $63.7_{1.8}$ | $73.9_{1.3}$ | $72.1_{0.4}$ | $47.5_{0.5}$ | $84.0_{1.4}$ |
| | 32 | $n$-shot | $75.3_{3.2}$ | $93.2_{1.9}$ | $77.7_{2.9}$ | $69.1_{1.9}$ | $58.3_{1.5}$ | $76.4_{2.2}$ | $74.2_{1.9}$ | $43.0_{1.5}$ | $84.5_{2.1}$ |
| | | STICL-S | $87.9_{0.6}$ | $97.9_{0.4}$ | $83.1_{0.9}$ | $74.0_{1.1}$ | $64.6_{1.2}$ | $79.4_{0.6}$ | $74.8_{1.5}$ | $56.5_{0.2}$ | $89.0_{0.4}$ |

Table 7: ID generalization scores of STICL-S for $n = 16$ shots and $|\mathcal{D}_{\text{unlab}}| = 200, 500$ for Llama 3 (8B). Results are shown for GLUE datasets with $n$-shot and STICL-S methods. The standard deviations of 10 runs are shown as subscripts.

| Model | $|\mathcal{D}_{\text{unlab}}|$ | GLUE | | | | | | |
|---|---|---|---|---|---|---|---|---|
| | | RTE | SST | QNLI | MNLI | COLA | MRPC | QQP |
| Llama 3 (8B) | 200 | $86.2_{0.4}$ | $97.2_{0.4}$ | $81.6_{1.0}$ | $73.9_{1.3}$ | $64.7_{1.1}$ | $78.9_{0.7}$ | $74.0_{0.5}$ |
| | 500 | $86.9_{0.3}$ | $97.1_{0.5}$ | $81.9_{0.7}$ | $74.8_{1.0}$ | $64.6_{0.8}$ | $81.4_{0.8}$ | $75.2_{0.3}$ |

Table 8: OOD generalization scores for Phi 3 and Llama 2 in a 16-shot scenario with $\mathcal{D}_{\text{unlab}} = 100$ over 10 runs with standard deviations shown as subscripts. In each dataset pair, demonstrations are taken from the left dataset, and the model is tested on the right dataset. The columns correspond to the results on the right datasets.

| Model | Method | QNLI $\rightarrow$ RTE | RTE $\rightarrow$ QNLI | QQP $\rightarrow$ MRPC | MRPC $\rightarrow$ QQP |
|---|---|---|---|---|---|
| Phi 3 (mini 4k) | $n$-shot | $64.3_{2.5}$ | $67.2_{1.5}$ | $63.7_{2.3}$ | $59.4_{2.2}$ |
| | PBFT | $64.1_{1.8}$ | $66.9_{1.6}$ | $64.7_{2.0}$ | $60.1_{1.4}$ |
| | STICL-S | $67.4_{0.6}$ | $69.2_{0.9}$ | $66.3_{2.4}$ | $64.4_{1.3}$ |
| Llama 2 (7B) | $n$-shot | $62.9_{2.3}$ | $66.3_{1.2}$ | $64.5_{1.9}$ | $61.1_{2.2}$ |
| | PBFT | $62.8_{1.3}$ | $68.1_{1.4}$ | $65.9_{1.8}$ | $61.3_{1.2}$ |
| | STICL-S | $64.8_{0.4}$ | $70.3_{0.6}$ | $67.8_{2.1}$ | $65.0_{1.1}$ |

## D.2 COMPARISON OF STICL AND METAICL

MetaICL (Min et al., 2022) shares conceptual similarities with STICL, as both methods aim to improve task generalization of ICL. However, the two approaches differ significantly in their training paradigms and mechanisms for handling task-specific information.

MetaICL updates the entire model through supervised fine-tuning across multiple tasks during meta-training, leveraging labeled data to condition the model on diverse task examples. This approach works well for smaller models, where full model fine-tuning is computationally feasible. However, MetaICL does not explicitly address latent shifts between demonstrations and queries, which can impact performance in certain settings.

In contrast, STICL employs a teacher-student framework within a self-training setup, where the teacher generates pseudo-labels for both demonstrations and queries. This enables task adaptation without additional labeled data, relying instead on unlabeled data for self-training. STICL updates only adapter modules, making it computationally efficient and scalable to larger models. Additionally, STICL explicitly disentangles latent shifts between demonstrations and queries, enhancing stability and generalization, particularly in OOD and low-resource scenarios.

We conducted experiments with MetaICL, adapting it to align with the STICL-S setup. When the same number of labeled instances was used, MetaICL effectively reduced to PBFT, where all labeled instances are combined into a single prompt. In contrast, STICL-S benefits from leveraging additional unlabeled data during its self-training phase. To address this difference, we modified MetaICL to include unlabeled instances with their true labels as part of its supervised fine-tuning process.

The experiments were conducted using Llama 3 (8B) under two configurations: 16 labeled and 100 unlabeled instances for STICL-S and 116 labeled instances for MetaICL. For MetaICL, we used batches of 16 labeled instances in individual prompts, requiring 8 iterations to fine-tune on all 116 instances. The results, averaged over 10 runs, are summarized in Table 9.

Table 9: Performance comparison of MetaICL and STICL-S across GLUE and MATH/MISC benchmarks.

| Method | RTE | SST | QNLI | MNLI | COLA | MRPC | QQP | MATH | MISC |
|--------|-----|-----|------|------|------|------|-----|------|------|
| MetaICL | 82.1 | 95.3 | 79.7 | 71.9 | 62.1 | 75.4 | 72.6 | 45.0 | 84.5 |
| STICL-S | 86.0 | 96.1 | 81.4 | 73.1 | 64.3 | 77.7 | 73.1 | 49.5 | 88.0 |

The results demonstrate that STICL-S consistently outperforms MetaICL across all datasets, even while utilizing fewer labeled instances during training. This improvement can be attributed to the weak-to-strong generalization mechanism in STICL-S, where the inclusion of additional unlabeled data enhances performance. Conversely, the marginal benefit observed from using more labeled data in MetaICL highlights the limitations of its supervised fine-tuning approach in this setup.

## D.3 FEW-SHOT STICL

STICL is primarily designed for 0-shot operation during the self-training phase, leveraging unlabeled data to encode task-specific information within the adapter. To examine its performance in few-shot setups, we evaluated STICL-S using Llama 3 (8B) in a 16-shot configuration, where 16 additional demonstrations were encoded into the adapter, resulting in a total of 32 labeled instances. This setup was compared against standard 32-shot ICL, as well as two STICL-S variants utilizing 32 labeled instances in a 0-shot configuration. Additionally, we included a baseline for a 0-shot setup with only 16 encoded demonstrations.

STICL is primarily designed to operate in a 0-shot setup during the self-training phase, leveraging unlabeled data to encode task-specific information in the adapter. To explore its performance in few-shot setups, we evaluated STICL-S with Llama 3 (8B) in a 16-shot configuration, where 16 additional demonstrations were encoded in the adapter, resulting in 32 labeled instances in total $(16+16)$. This configuration was compared to standard 32-shot ICL, as well as two STICL-S variants that use 32 labeled instances in a 0-shot setup. Additionally, we included results for a 0-shot configuration where only 16 demonstrations were encoded in the adapter.

To standardize comparisons, we denote each STICL variant using the format $n/d$, where $n$ represents the number of shots (n-shot) and $d$ indicates the number of demonstrations encoded in the adapter. The results, averaged over 10 runs, are shown in Table 10.

Table 10: Performance comparison of STICL-S configurations and standard 32-shot ICL averaged over 10 runs.

| Method | RTE | SST | QNLI | MNLI | COLA | MRPC | QQP | MATH | MISC |
|---|---|---|---|---|---|---|---|---|---|
| 32-shot ICL | 75.3 | 93.2 | 77.7 | 69.1 | 58.3 | 76.4 | 74.2 | 43.0 | 84.5 |
| STICL-S (0/32) | 87.9 | 97.9 | 83.1 | 74.0 | 64.6 | 79.4 | 74.8 | 56.5 | 89.0 |
| STICL-S (0/16) | 86.0 | 96.1 | 81.4 | 73.1 | 64.3 | 77.7 | 73.1 | 49.5 | 88.0 |
| **STICL-S (16/16)** | 87.3 | 96.4 | 82.2 | 74.6 | 65.4 | 78.2 | 74.5 | 51.0 | 89.0 |

The results demonstrate that STICL-S in the 16-shot configuration with 16 encoded demonstrations (16/16) outperforms both standard 32-shot ICL and STICL-S (0/16) across all datasets, showcasing its ability to utilize additional context during inference. However, it slightly underperforms compared to the 0-shot STICL-S variant with 32 encoded demonstrations (0/32), likely due to the self-training process that is exclusive to the 0-shot setup. Nevertheless, the strong performance in n-shot setups ($n > 0$) highlights the flexibility and efficacy of STICL-S in leveraging additional context provided within the prompt.

### D.4 FAITHFUL ENCODING AND RETRIEVAL OF DEMONSTRATIONS

To evaluate whether demonstrations are faithfully encoded and disentangled, we conducted an experiment by encoding a single demonstration into the adapter and assessing the student model's ability to capture this information. Specifically, we utilized 1000 examples per dataset across the GLUE benchmark using Llama 3 (8B).

For each dataset, the student model was prompted with a simple instruction: "Repeat the demonstration word for word." During the self-training phase, the teacher model processed input examples using the following template: "Demonstration: {*demonstration*}. Answer: ({*answer*})." The adapter learned to encode demonstration-specific information indirectly by aligning its outputs with the teacher's responses, without explicitly seeing the demonstration itself. After training, the similarity between the student model's response and the original demonstration was computed. Table 11 shows the average BERTScore similarity (Zhang et al., 2020) between the original demonstrations and the student's reconstructed response.

Table 11: Average BERTScore ($F_1$) similarity across GLUE datasets. Higher scores indicate better fidelity in recalling the encoded demonstration.

| | RTE | SST | QNLI | MNLI | COLA | MRPC | QQP |
|---|---|---|---|---|---|---|---|
| BERTScore | 0.84 | 0.91 | 0.80 | 0.83 | 0.86 | 0.82 | 0.81 |

The consistently high BERTScore values across all datasets indicate that the student model can reliably retrieve the encoded demonstration from the adapter. This suggests that STICL effectively disentangles and stores task-specific information within the adapter's weights. Notably, when compared to standard ICL, STICL often produced different outputs for certain queries, particularly in instances where it corrected "corrupted" labels provided by the teacher. Despite these differences, the student model maintained a high degree of semantic similarity in reproducing the demonstrations. This suggests that the adapter weights capture not only the demonstration itself but also additional latent information that contributes to improved generalization.

We present below a pair of examples from SST and RTE, chosen to represent reconstructed demonstrations with similarity scores close to the dataset averages.

**SST examples**

- **Example 1**
  - **Original:** *Proves once again he hasn't lost his touch, delivering a superb performance in an admittedly middling film.*
    *Answer:* (Positive)

- **Reconstructed:** *He demonstrates once more that he hasn't missed a beat, delivering a remarkable performance in what is admittedly an average film.* *Answer:* (Positive)

- **Example 2**

    - **Original:** *Though many of the actors spark briefly when they first appear, they can't generate enough heat in this cold vacuum of a comedy to ignite a reaction.*
      *Answer:* (Negative)

    - **Reconstructed:** *Although some actors manage to show a hint of energy early on, they fail to create any real warmth or spark within this lifeless and chilly comedy.* *Answer:* (Negative)

**RTE examples**

- **Example 1**

    - **Original:** Premise: *The source added that the investigation proved that the bases of the genocide crime "were completed with a series of illegal arrests followed in some cases with assassinations or cases of disappearances and were preceded, according to information attached to the file, by cases of torture."*
      Hypothesis: *Investigators discovered that a series of illicit arrests were often followed by disappearances or murders and were preceded by torture.*
      *Answer:* (True)

    - *Reconstructed:* Premise: *The investigation confirmed that genocide involved illegal arrests followed by disappearances or murders, often preceded by torture. Hypothesis: Investigators found that unlawful arrests frequently resulted in disappearances or murders, often preceded by acts of torture. Answer:* (True)

- **Example 2**

    - **Original:** Premise: *American tobacco companies were showing a profit most quarters due to export sales of cigarettes and diversification of products sold, including food.*
      Hypothesis: *PM often entered markets with both cigarettes and food.*
      *Answer:* (False)

    - **Reconstructed:** Premise: *Profitability was often maintained by American tobacco companies through diversification into food products and successful cigarette exports.*
      Hypothesis: *Philip Morris International offered food items and cigarettes. Answer:* (False)

# E  EXPERIMENTAL DETAILS

## E.1  MODELS

For all three models – Llama 3, Llama 2, and Phi 3 – we utilize the `bfloat16` half-precision format for parameters. A summary of the models is provided in Table 12.

## E.2  HYPERPARAMETERS

We employ the AdamW optimizer (Loshchilov & Hutter, 2019) for both PBFT and STICL variants, with a learning rate of $10^{-4}$. For ICV (Liu et al., 2023) and Batch-ICL (Zhang et al., 2024), we follow the implementations provided in the original papers and adapt them to our codebase, using their default parameters where specified. In the case of Batch-ICL, we utilize attention heads from the last 20 layers ($k = 20$) and fine-tune the model for 10 epochs.

**LoRA adapter configuration.**

- **r** $= 8$
  The rank of the low-rank matrices used to decompose the original weight matrix in LoRA. A smaller $r$ reduces the parameter count while retaining essential information.

- **α = 32:**
  A scaling factor applied to the low-rank updates, balancing the influence of the original weights and the low-rank matrices.
- **Dropout:** 0.1
  The dropout rate applied to the low-rank updates.
- **Target modules:**
  `q_proj, k_proj, v_proj, o_proj, gate_proj, up_proj, down_proj`

### E.3 COMPUTING INFRASTRUCTURE

We conducted our experiments on *AMD Ryzen Threadripper 3970X 32-Core Processors* and 4× *NVIDIA GeForce RTX 3090* GPUs with 24GB of RAM.

Table 12: Summary of the models used in the experiments, including their Hugging Face IDs, parameter counts, context window sizes, training token volumes, and adapter sizes.

| Model | Hugging Face ID | Parameters | Context window size | Training tokens | Adapter size |
|---|---|---|---|---|---|
| Llama 3 | Meta-Llama-3-8Bb | 8B | 8k | 15T | 21M |
| Llama 2 | Llama-2-7b | 7B | 4k | 2T | 20M |
| Phi 3 | Phi-3-mini-4k-instruct | 3.8B | 4k | 3.3T | 4.5M |

## F PROMPT TEMPLATES

### F.1 GLUE PROMPT STRUCTURE

---

**Generic prompt template for GLUE tasks**

**Demonstrations:**

```
{Sentence 1}
{Sentence 2 (if applicable)}
Answer: ({Correct answer})
```
**Query:**
```
{Sentence 1}
{Sentence 2 (if applicable)}
Question: {Task-specific question}
Answer: (
```

---

The prompts for GLUE tasks typically consist of two sentences (or one in certain cases) followed by a task-specific question and the corresponding answer. The model is expected to choose from predefined labels like *Yes/No*, *True/False*, or specific class names based on the dataset. The phrasing of the question preceding each answer in the demonstrations is specific to the task. Below is a list of the questions used for each GLUE dataset. To encourage the model to select from predefined labels, we prepend the phrase "answer with one word" before each question, and we append clarifying options such as *Yes or No?* to prompt a more targeted response:

- **RTE**: `{hypothesis} True or False?`
- **SST**: `What is the sentiment? Positive or Negative?`
- **QNLI**: `Does the sentence answer the question? Yes or No?`
- **MNLI**: `Is the second sentence an Entailment, Contradiction, or Neutral?`
- **COLA**: `Is this sentence linguistically acceptable? Yes or No?`
- **MRPC**: `Do both sentences say the same thing? Yes or No?`
- **QQP**: `Do both questions ask the same thing? Yes or No?`

## F.2 MMLU PROMPT STRUCTURE

---

**Generic prompt template for MMLU sub-datasets**

**Demonstrations:**

```
Question: {Previous Question 1}
Answer choices:
  (A: {Choice A1}),
  (B: {Choice B1}),
  (C: {Choice C1}),
  (D: {Choice D1})
Answer: (Correct Answer 1)

Question: {Previous Question 2}
Answer choices:
(A: {Choice A2}),
(B: {Choice B2}),
(C: {Choice C2}),
(D: {Choice D2})
Answer: (Correct Answer 2)
...
```

**Query:**

```
Question: {Current Question}
Answer choices:
(A: {Choice A}),
(B: {Choice B}),
(C: {Choice C}),
(D: {Choice D})
Answer: (
```

---

**Example for MMLU `elementary_math` (MATH)**

**Demonstrations:**

```
Question: Ms. Perez drove a total of 40 miles in 5 days.
She drove the same number of miles each day.
How many miles did Ms. Perez drive each day?
Answer choices: (A: 5), (B: 7), (C: 8), (D: 9)
Answer: (C: 8)

Question: Find the median in the set of data
23, 13, 18, 29, 32, 25.
Answer choices: (A: 18), (B: 24), (C: 25), (D: 29)
Answer: (B: 24)
```

**Query:**

```
Q: A worker on an assembly line takes 7 hours to produce
22 parts. At that rate how many parts can she produce
in 35 hours?
Answer choices:
(A: 220 parts),
(B: 770 parts),
(C: 4 parts),
(D: 110 parts)
Answer: (
```

---