# OpenReview forum: "Disentangling Latent Shifts of In-Context Learning Through Self-Training"
_ICLR.cc/2025/Conference — Submitted to ICLR 2025_

### Official Review · Reviewer_eung · 2024-10-30

**Soundness:** 3
**Presentation:** 3
**Contribution:** 3
**Rating:** 8
**Confidence:** 3

**Summary:**

When using in-context learning to solve tasks with LLMs, they can suffer from instability, characterised by biases toward the first and last examples in the demonstrations. Furthermore, using many demonstrations in long contexts extends the length of the LLM’s prefix window, increasing computation. This paper proposes a method to disentangle unlabelled demonstrations in-context from queries using self-training. A teacher model sees the demonstrations along with the query and produces a pseudo-label that a student model is trained on using adapters. These adapters (approximately theoretically) learn the demonstrations, allowing the student to be prompted with only the query and achieve comparable (in fact, better due to a weak-to-strong generalisation effect) performance than n-shot prompting. The method is also shown to outperform other comparable approaches.

**Strengths:**

As far as I am aware, the method (self-training from unlabelled demonstrations using a teacher to generate pseudo-labels) is simple but novel. However, I note that I am not extensively familiar with the literature in this area. The paper is clearly written and mostly easy to understand. The idea is nicely motivated by the theory used based on linear attention. I think that the work is significant. Long-context models suffer from serious inference-time bottlenecks. This work has not only shown a way to eliminate this bottleneck in certain scenarios where demonstrations may dominate (which on its own I would deem significant), but also improves in terms of quality compared to comparable approaches. The discussion on weak-to-strong generalisation is interesting and valuable.

**Weaknesses:**

- There is no section on limitations. I would be interested in seeing some discussion on this with respect to both the chosen method and results. In particular, the computational cost of training vs using demonstrations, when is the method not applicable, and the lack of exploration with very large demonstration sets.
- Some of the discussion on weak-to-strong generalisation in section 2 is unclear. It does not explain why the student is expected to outperform the teacher or about what local-to-global consistency is in the context of in-context learning. I had to refer quite a lot to the cited paper to understand what was being said here. Please clarify.
- The theory is interesting motivation but based on an approximation of quadratic attention. To provide further support, given the limited theory, is there a way you could experimentally determine whether the demonstrations have been disentangled?

**Questions:**

- Regarding the evaluation setup, are the queries used in training the student with pseudo-labels disjoint from those used when evaluating the student?
- Is stability an issue beyond transformer-based LLMs? E.g. for state-space/diffusion-based LLMs?
- Do you have any quantification of how much more efficient your method has made inference?

---

> ### Author Response · Authors · 2024-11-20
>
> Thank you for your thoughtful comments. We're glad you found our method straightforward, significant, and clearly presented. We appreciate your recognition of self-training with pseudo-labels as a way to address inference-time bottlenecks and improve performance.
>
> ---
>
> > There is no section on limitations. I would be interested in seeing some discussion on this with respect to both the chosen method and results. ...
>
> Thank you for raising this point. Below are key limitations of STICL, which will be included in the revised manuscript.
>
> 1. **Computational cost**:
> STICL introduces extra overhead due to adapter fine-tuning during self-training. While lighter than full model fine-tuning, it is still costlier than standard ICL, which avoids weight updates. However, STICL offsets this by removing demonstrations from the input during inference. For instance, processing 16 demonstrations with Llama 3 (8B) takes ~120x longer than a 0-shot setup due to the self-attention bottleneck. Based on our measurements, self-training with 100 unlabeled instances and 16 demonstrations has a computational cost equivalent to ~2100 inferences in a 16-shot setup, after which the fine-tuning cost is balanced out.
>
> 2. **Applicability**:
> STICL relies on unlabeled data, making it less suitable for extremely resource-limited settings. In our experiments, 100 unlabeled instances with up to 32 demonstrations sufficed for strong performance. However, acquiring even modest amounts of unlabeled data could be challenging in some cases.
>
> 3. **Large demonstration sets**:
> Although STICL efficiently encodes demonstrations into adapters to overcome context length limitations, the method has not been extensively tested with very large demonstration sets. Our findings suggest that using multiple adapters with manageable demonstration sizes is more effective as the number of demonstrations increases. While this approach could theoretically scale further, its effectiveness with much larger sets remains untested, and additional adapters increase computational costs, creating a tradeoff between scalability and efficiency.
>
> ---
>
> > Some of the discussion on weak-to-strong generalisation in section 2 is unclear. It does not explain why the student is expected to outperform the teacher or about what local-to-global consistency is in the context of in-context learning.
>
> Thank you for pointing this out. We will clarify these concepts in the revised manuscript.
>
> The concept of weak-to-strong generalization in Section 2 assumes the student model leverages data structure and robustness to local perturbations to improve beyond the teacher's pseudo-labels. While the teacher relies on ICL with fixed demonstrations, the student benefits from fine-tuning, refining noisy pseudo-labels to align with true labels, and learning consistent representations. Local-to-global consistency enables the student to maintain stable outputs under small input changes (local consistency) and generalize these patterns to new or complex queries (global consistency). Prior work (e.g., Lang et al., 2024) establishes theoretical bounds on the student's performance, showing that its error is directly proportional to pseudo-label error and inversely proportional to robustness, which we approximate using the Lipschitz constant.
>
> ---
>
> > … is there a way you could experimentally determine whether the demonstrations have been disentangled?
>
> Thank you for this question. This prompted us to validate further STICL's ability to disentangle shifts, where we designed an experiment to assess how faithfully encoded demonstrations are captured in the adapter's weights. Using 1000 examples per GLUE dataset with Llama 3 (8B), we encoded a single demonstration and assessed the student model's ability to reproduce it by calculating the BERTScore (F1) similarity between the student's output and the original demonstration. The averaged results are shown below:
>
> | **Metric**                | RTE | SST | QNLI | MNLI | CoLA | MRPC | QQP |
> |---------------------------|---------|---------|----------|----------|----------|----------|---------|
> | BERTScore | 0.84    | 0.91    | 0.80     | 0.83     | 0.86     | 0.82     | 0.81    |
>
>
> The consistently high BERTScore values demonstrate that the student model faithfully retrieves the encoded demonstration from the adapter.
>
> ---
>
> ### Questions
>
> > Q1
>
> Yes, the queries used for training the student are drawn from the training set, with their labels removed to simulate an unlabeled dataset. For evaluation, queries are exclusively drawn from the test set.
>
> ---
>
> > Q2
>
> To the best of our knowledge, stability in ICL is not limited to transformer-based models. State-space models may face numerical stability challenges and prompt drift, diffusion models risk compounded errors during iterative denoising, and RNNs often exhibit sensitivity to sequence length and order dependence.
>
> ---
>
> > Q3
>
> Please refer to our earlier response on the limitations.

---

> > ### Comment · Reviewer_eung · 2024-11-21
> > **Thanks to Authors**
> >
> > Thanks very much for your responses and for answering my questions.
> >
> > Regarding the additional experiment about empirically validating disentanglement of demonstrations. How were you prompting the Llama model to produce the encoded demonstration? Could you explain this experiment in more detail?

---

> > > ### Author Response · Authors · 2024-11-22
> > >
> > > Of course, we hope that the following explanation provides clarity. To evaluate how faithfully the demonstrations were encoded, we prompted the student model (with the adapter activated) with a simple instruction: "Repeat the demonstration word for word." During the self-training phase, the teacher model processed input examples using our standard template: "Demonstration: {`demonstration`}. Answer: ({`answer`})." The adapter learned to encode demonstration-specific information indirectly by aligning its outputs with the teacher's responses, without explicitly seeing the demonstration itself. After training, we computed the similarity between the student model's response to the original demonstration.
> > >
> > > When comparing STICL with standard ICL, STICL often produced different outputs for some queries, particularly in cases where it corrected the "corrupted" labels provided by the teacher. Despite these differences, the student model consistently reproduced the demonstrations with a high degree of semantic similarity, as measured by BERTScore. This suggests that the adapter weights store not only the encoded demonstration but also additional latent information that enhances generalization. We are uncertain whether this latent information can be explicitly expressed in textual form, but it is an intriguing possibility that we are eager to explore in future research.

---

> > > > ### Comment · Reviewer_eung · 2024-11-22
> > > >
> > > > Thanks for the explanation. I now understand this experiment. BERTScore can be quite brittle. Could you also provide some samples of the generated demonstrations? I would be curious to see these.

---

> > > > > ### Author Response · Authors · 2024-11-22
> > > > >
> > > > > Here are a few examples from SST and RTE, chosen to represent reconstructed demonstrations with similarity scores near the dataset's average.
> > > > >
> > > > > ---
> > > > >
> > > > > ## SST
> > > > >
> > > > > ### Example 1
> > > > > Original:
> > > > > -  Proves once again he hasn't lost his touch, delivering a superb performance in an admittedly middling film.\
> > > > >    Answer: (Positive)
> > > > >
> > > > > Reconstructed:
> > > > > - He demonstrates once more that he hasn't missed a beat, delivering a remarkable performance in what is admittedly an average film. Answer: (Positive)
> > > > >
> > > > > ### Example 2
> > > > > Original:
> > > > > -  Though many of the actors spark briefly when they first appear, they can't generate enough heat in this cold vacuum of a comedy to ignite a reaction.\
> > > > > Answer: (Negative)
> > > > >
> > > > > Reconstructed:
> > > > > - Although some actors manage to show a hint of energy early on, they fail to create any real warmth or spark within this lifeless and chilly comedy. Answer: (Negative)
> > > > > ---
> > > > >
> > > > > ## RTE
> > > > >
> > > > > ### Example 1
> > > > >
> > > > > Original:
> > > > > - **Premise:** The source added that the investigation proved that the bases of the genocide crime "were completed with a series of illegal arrests followed in some cases with assassinations or cases of disappearances and were preceded, according to information attached to the file, by cases of torture."
> > > > > **Hypothesis:** Investigators discovered that a series of illicit arrests were often followed by disappearances or murders and were preceded by torture.\
> > > > > Answer: (True)
> > > > >
> > > > > Reconstructed:
> > > > > -  **Premise:** The investigation confirmed that genocide involved illegal arrests followed by disappearances or murders, often preceded by torture. **Hypothesis:** Investigators found that unlawful arrests frequently resulted in disappearances or murders, often preceded by acts of torture. Answer: (True)
> > > > >
> > > > >
> > > > > ### Example 2
> > > > >
> > > > > Original:
> > > > > - **Premise:** American tobacco companies were showing a profit most quarters due to export sales of cigarettes and diversification of products sold, including food.
> > > > > **Hypothesis:** PM often entered markets with both cigarettes and food.\
> > > > > Answer: (False)
> > > > >
> > > > > Reconstructed:
> > > > > - **Premise:** Profitability was often maintained by American tobacco companies through diversification into food products and successful cigarette exports.  **Hypothesis:** Philip Morris International offered food items and cigarettes.  Answer: (False)
> > > > >
> > > > > ---
> > > > >
> > > > > If you're interested in seeing more examples, please let us know — we'd be happy to share them.

---

> > > > > > ### Comment · Reviewer_eung · 2024-11-22
> > > > > >
> > > > > > Thanks for providing this so quickly! Great work.

---

> > > > > > > ### Author Response · Authors · 2024-11-22
> > > > > > >
> > > > > > > Thank you, we truly appreciate it! We're also grateful for your question, as it led us to uncover some intriguing insights.
> > > > > > >
> > > > > > > We'd like to clarify that during self-training, the cross-entropy loss considers all logits, not just the values corresponding to the verbalizers (words representing the classes). While this is outlined in equation (7) of the manuscript, we will emphasize it further in the text in light of these findings.
> > > > > > >
> > > > > > > Moreover, your question inspired us to explore a new direction for future work: evaluating how effectively demonstrations can be reconstructed when the teacher and student models are based on different architectures. We believe this approach offers a great opportunity to assess the functional compatibility between distinct architectures.

---

### Official Review · Reviewer_sdep · 2024-11-04

**Soundness:** 3
**Presentation:** 3
**Contribution:** 4
**Rating:** 3
**Confidence:** 4

**Summary:**

The paper collects two issues of ICL: (1) instability: ICL is affected by ordering and selection of demonstrations (2) context length: ICL is limited by context window.
The paper assumes that the demonstration of ICL introduces a latent shift to the language model.
The paper points out a series of works on linear functions have a similar idea that demonstrations shift zero-shot $W_\text{zs}$ to $W_\text{zs}+\Delta W_\text{ICL}$, where $\Delta W_\text{ICL}$ can be regarded as the latent shift coming from demonstrations.
The paper then introduces existing methods on disentangling latent shift and the main idea of the paper, i.e., the proposed method of finetuning an LLM with query aligning its output to an LLM with both demonstrations and query.
Therefore, the latent shift is injected into the LLM via alignment, and when inference, there is no demonstration needed.
Finally, through experiments, the paper shows that the proposed method can improve stability and generalization, and show weak-to-strong generalization.
The paper also mentioned the proposed method is more efficient than vanilla ICL since no demonstration is involved in the input prompt.

**Strengths:**

(1) The proposed method is straightforward, makes sense, and very interesting -- regarding ICL as an alignment task is very interesting.

(2) The proposed method can handle the ordering issue since in the alignment state, the input of the teacher model can take input with arbitrary order. It's not very clear how to handle the ordering issue in vanilla ICL, but in the proposed method, the ordering issue is handled naturally via training sample construction (STICL-S in Table 2).

(3) The proposed method can reduce the inference cost, since no demonstration is involved when inference.

(4) The proposed method has better scores than baselines for ID generalization (Table 1).

(5) The proposed method has better scores than baselines for OOD generalization (Table 3)

**Weaknesses:**

(1) Though the proposed method decreases the inference of cost since no demonstration is involved, it requires training for alignment.

(2) It's unclear how the paper posits Sec. 3.2 since the existence of STICL-R. It's more natural for me to take 16 samples from k$\times$16 examples to construct 16-shot demonstrations for STICL-R, rather than construct k separate 16-shot sets for k adapters in STICL-S. The author could consider clarifying their reasoning behind the design choices for STICL-R and STICL-S

(3) It's unclear how the paper calculates the Lipschitz constant for each method in Figure 2, due to page limitation. The author could consider providing a more detailed explanation of their Lipschitz constant calculation method in the Appendix.

(4) It's unclear how the paper desgins the pseudo-label correction, due to page limitation. The author could consider providing a step-by-step description of their pseudo-label correction algorithm, possibly with package algorithm.

**Questions:**

I like the idea very much, but some things are unclear in the paper, thus I lower my score.

(1) This is an important one. What is the algorithm for the pseudo-label correction? I can only find several descriptions for it. Could the author give a detailed description of it?

(2) This is less important. How is the Lipschitz constant calculated? I still can only find some rough description of it in the appendix on Page 17, but without a detailed description.

(3) It would be interesting to see the scores of 16-shot on the student model. We know the student is good at 0-shot, how about 16-shot?

Others:

(4) The idea of Sec. 3.2 could be applied to joint training of multiple tasks. In Table 3, we see OOD generalization via training on task A and testing on task B. It would be interesting to train multiple tasks and test on the others.

---

> ### Author Response · Authors · 2024-11-20
>
> Thank you for your valuable comments and suggestions, especially for proposing ideas for expanding our method to other setups. We are glad that you find our paper clear and interesting. Thank you for recognizing the potential of our method to mitigate the demonstration ordering issues, reduce inference cost, and improve ID and OOD generalization.
>
> ---
>
> > It's more natural for me to take 16 samples from k 16 examples to construct 16-shot demonstrations for STICL-R, rather than construct k separate 16-shot sets for k adapters in STICL-S.
>
> Thank you for raising this point. The motivation behind STICL-R is to provide a comparison with STICL-S, which we propose as the superior and more practical approach. STICL-R requires more labeled data, as it resamples demonstrations in each iteration, yet it is still outperformed by STICL-S. Our experiments aimed to validate the intuition that resampling can degrade performance, likely due to overloading the adapter with too many demonstrations over the course of multiple epochs. We actually tested the proposed approach using k different 16-shot sets with a single adapter. However, this approach yielded limited success. We found that dedicating a separate adapter to each set resulted in improved performance. The multiple adapters can be merged into a single adapter for inference, ensuring efficiency while leveraging the benefits of multiple adapters during training.
>
> ---
>
> ### Questions
>
> > Q1
>
> Pseudo-label correction is the process where the student model, in a teacher-student framework, refines noisy pseudo-labels generated by the teacher, aligning them closer to true labels. This correction links the student’s true error to its error on pseudo-labeled data, leveraging the structure of the data. It assumes two types of points:
> -   **"Good" points** with correct pseudo-labels.
> -   **"Bad" points** with incorrect pseudo-labels.
>
> Bad points are assumed to be surrounded by good points in the feature space ("neighborhoods"). If the model is robust to small perturbations, it can use good points to correct bad ones.
>
> A formal bound can be derived (Lang et al., 2024) to show that the true error of the student model depends on:
> 1.  The rate of pseudo-label errors.
> 2.  The robustness of the classifier in local neighborhoods.
> 3.  The proportion of bad points that are close to good points.
>
> If these conditions are met, pseudo-label correction improves performance beyond learning from noisy labels. We will revise the manuscript to clarify this phenomenon.
>
> ---
>
> > Q2
>
> We approximate the Lipschitz constant by computing the Frobenius norm of the input-output Jacobian matrix of the network. In Appendix B, we lay out how this approximation connects the Jacobian matrix to the Lipschitz constant. Specifically, for a given method, we first complete the training phase. Afterward, we compute the Jacobian matrices for selected instances, using the embeddings as input variables and the penultimate layer activations as output variables. We will revise our manuscript to clarify this process.
>
> ---
>
> > Q3
>
> Thank you for this interesting question! We additionally evaluated STICL-S with Llama 3 (8B) in a 16-shot setup with 16 additional demonstrations encoded in the adapter (32 labeled instances in total). We compared this configuration to the standard 32-shot setup and two 0-shot STICL-S variants using 16 and 32 labeled instances, respectively. In the table below, we denote the STICL variant using the format **n/d**, where **n** represents the number of shots (n-shot) and **d** indicates the number of demonstrations encoded in each adapter. The results (averaged over 10 runs) are as follows:
>
> | Method   | RTE    | SST    | QNLI   | MNLI   | COLA   | MRPC   | QQP    | MATH   | MISC   |
> |----------|--------|--------|--------|--------|--------|--------|--------|--------|--------|
> | 32-shot (standard ICL)  | 75.3   | 93.2   | 77.7   | 69.1   | 58.3   | 76.4   | 74.2   | 43.0   | 84.5   |
> | STICL-S (0/32) | 87.9   | 97.9   | 83.1   | 74.0  | 64.6   | 79.4   | 74.8   | 56.5   | 89.0   |
> | STICL-S (0/16) | 86.0 | 96.1 | 81.4 | 73.1 | 64.3 | 77.7 | 73.1 | 49.5 | 88.0 |
> | **STICL-S (16/16)** | 87.3   | 96.4   | 82.2   | 74.6   | 65.4   | 78.2   | 74.5   | 51.0   | 89.0   |
>
>
> STICL-S in the 16-shot setup with 16 demonstrations encoded outperforms standard 32-shot ICL and STICL-S (0/16) across all datasets, leveraging additional context during inference. However, it performs slightly worse than 0-shot STICL-S variants with 32 labeled instances (0/32), likely due to the self-training process exclusive to 0-shot setups. Nonetheless, the model performs well in n-shot setups where n > 0.
>
> ---
>
> > Q4
>
> Thank you for this valuable suggestion. We reflected on this idea and concluded that exploring joint training of multiple tasks and analyzing its impact on OOD generalization is indeed a promising direction, which we plan to explore in future work.

---

> > ### Comment · Reviewer_sdep · 2024-11-21
> > **Ack (score 6->8)**
> >
> > Thank you for addressing my questions. The new table is helpful. I raised the score.

---

> > > ### Author Response · Authors · 2024-11-22
> > >
> > > Thank you!

---

### Official Review · Reviewer_gSE5 · 2024-11-10

**Soundness:** 3
**Presentation:** 2
**Contribution:** 3
**Rating:** 6
**Confidence:** 3

**Summary:**

The paper proposes to use unlabeled data to stabilize and improve in-context learning. The paper formalizes a form of decomposition of the information obtained from demonstrations vs query. The particular approach involves utilizing pseudolabels in a teacher-student framework. Empirical results suggest that this form of self-training can effectively leverage labeled data to improve robustness and stability in downstream performance.

**Strengths:**

The self-training analysis of the paper is interesting and useful - connecting to "classical" ideas of local consistency and coverage expansion. This work brings in the perspective of self-training to LLMs and shows that with just a few demonstrations but large amount of unlabeled data, one can adapt LLMs to have strong and robust performance on various downstream tasks.

The approach, while adding complexity over vanilla ICL, is fairly intuitive and builds on solid foundations of how pseudolabels can enable weak-to-strong generalization and can be leveraged for better and more robust performance. This is an important question of real-world significance.

The empirical investigations span a variety of datasets and models.

**Weaknesses:**

One main weakness (and source of confusion for me) is the terminology of "in-context learning" when the models are being fine-tuned via self-training. The whole point of ICL is to eliminate the need for finetuning models, so it is confusing how the current approach fits into the framework. The current process does some kind of finetuning, so it should be pitched as a finetuning method. The method is much more expensive than n-shot, and in general, the computational aspects need to be considered for fair comparisons.

Keeping terminology aside, the motivation of the paper - in terms of disentangling latent shifts is also quite confusing and seems unnecessary. It seems like the goal is to just "distill" the ICL process into the adapter. I'd recommend the authors to simplify this presentation a bit, because it is currently distracting from the main message. I don't think this disentanglement introduces a fundamentally new perspective - in-context learning can be approximated by (the more expensive) finetuning on related task data. As far as I understand, that is the main connection.

**Questions:**

(1) For the pattern-based finetuning baseline - do you finetune on just the {4, 8, 16, 32} demonstrations?

(2) Are the demonstrations used at train and test-time the same or different? How should one think of what the demonstrations are meant

(3) I'm assuming the "unlabeled samples" come from the standard training set, with labels removed? What's the "oracle" performance of finetuning on ground truth labels?

(4) Is the following a reasonable story that faithfully captures what's going on? The main idea is to do some form of self-training (via finetuning an adapter), and the labels are generated on this unlabeled data via in-context learning?

---

> ### Author Response · Authors · 2024-11-20
>
> Thank you for your thoughtful comments and suggestions. We appreciate that you found the self-training analysis interesting and useful, and that you recognize the practical significance of our work.
>
> ---
>
> > One main weakness (and source of confusion for me) is the terminology of "in-context learning" when the models are being fine-tuned via self-training. … The method is much more expensive than n-shot, and in general, the computational aspects need to be considered for fair comparisons.
>
> Thank you for raising this point. While STICL incorporates fine-tuning through self-training, its primary goal remains to enhance ICL. Crucially, ICL is an integral part of STICL, as the teacher model relies on ICL to generate pseudo-labels. In this framework, fine-tuning is used specifically to align the student model's outputs with the teacher's pseudo-labels. For this reason, we decided to keep "ICL" in the method's name, as the approach principally relies on ICL. Although one of the key advantages of ICL over fine-tuning is that it eliminates the need for weight updates, ICL has significant limitations. As we noted in the Introduction, ICL struggles with effectively leveraging information from long contexts and is constrained by the context window size, limiting its ability to incorporate multiple demonstrations. STICL addresses these drawbacks by combining ICL with lightweight fine-tuning of adapter modules, overcoming the context size limitation while preserving the foundational principles of ICL.
>
> Regarding computational costs, although fine-tuning is generally more expensive than n-shot ICL, the fine-tuning of adapters in STICL is computationally lightweight. It can also offset some of its cost in practice by eliminating the need for repeated demonstrations during inference, effectively enabling a 0-shot setup. Prompted by your remark, we conducted additional experiments to compare the computational costs of n-shot ICL and STICL. For example, with Llama 3 (8B) processing 16 demonstrations from GLUE datasets, inference takes approximately 120 times longer than a 0-shot setup (processing only the query). Additionally, we calculated that in this setup, the computational cost of self-training with a single adapter is roughly equivalent to performing 2100 inferences in a 16-shot setup. This means that after about 2100 inferences, the time invested in fine-tuning is effectively offset by the efficiency gains during inference. We will include this analysis in the revised manuscript.
>
> ---
>
> > ... I don't think this disentanglement introduces a fundamentally new perspective - in-context learning can be approximated by (the more expensive) finetuning on related task data. …
>
> We understand your suggestion that ICL can be approximated as a form of fine-tuning, a perspective we agree with. However, our contribution is not focused on introducing a new framing of ICL. Instead, we view our work as a practical exploration of disentangling latent shifts to enhance and simplify the ICL process, drawing inspiration from similar approaches in the literature (Hendel et al., 2023; Liu et al., 2023). One concrete benefit of this disentanglement is the elimination of demonstrations from the input context, which effectively shortens the prompt and addresses one of ICL's key limitations: the context window size.
>
> Disentangling latent shifts not only improves performance but also provides insights into the internal mechanisms of ICL, potentially aiding model interpretability. While this is not our primary focus, similar ideas have been explored, such as Todd et al. (2024)'s use of function vectors to represent and extract task functions from LLMs. Our work complements these studies by offering a practical approach to enhancing ICL and contributing to model interpretability.
>
> ---
>
> ### Questions
>
>
> > Q1
>
> Yes, we fine-tune using the same number of labeled instances as in the other methods to ensure a fair comparison across approaches.
>
> ---
>
> > Q2
>
> For STICL, demonstrations are labeled instances drawn from the training set and are used by the teacher model to generate pseudo-labels for queries, which are sampled from the unlabeled set. During test time, the information from these demonstrations is effectively encoded in the adapter, so they are not used as part of the input context. Instead, queries from the test set are directly used to evaluate the models, with the adapter providing the necessary task-specific context.
>
> ---
>
> > Q3
>
> Exactly, the "unlabeled samples" are drawn from the training set with their labels removed. Using ground truth labels instead of pseudo-labels to fine-tune the student model improves performance, though it would require labeled data in practice. This supports the hypothesis that pseudo-label correction is key to weak-to-strong generalization, as STICL’s student model refines corrupted pseudo-labels to better align with ground truth labels.
>
> ---
>
> > Q4
>
> Yes, we believe this captures the main idea of STICL.

---

> ### Author Response · Authors · 2024-11-25
>
> We'd like to kindly remind you about our responses to your review. If there's anything further you'd like us to clarify, we'd be happy to provide additional details.

---

> > ### Comment · Reviewer_gSE5 · 2024-12-01
> > **Response to rebuttal**
> >
> > Dear authors,
> >
> > Thank you for your response. I agree with the responses and I continue to support the acceptance of the paper. However, I strongly believe the manuscript should incorporate the following changes:
> >
> > - Make it clear that "ST-ICL" is not a form of in-context learning but rather a lightweight finetuning method that incorporates in-context learning. I believe in-context learning is a privileged term reserved for no parameter updates, and the present terminology is confusing and can be misleading.
> >
> > - I also strongly suggest that the authors de-emphasize the perspective on disentanglement, and present the method, the benefits and tradeoffs more directly. That would strengthen the paper.
> >
> > In light of my comments above, I keep my score and recommend acceptance.
> >
> > Thanks!

---

> ### Author Response · Authors · 2024-12-03
>
> Thank you for your additional feedback and support for the acceptance of our paper. We appreciate the time and effort you’ve taken to engage with our work and provide constructive suggestions. We would like to take this opportunity to further clarify our views on the points you raised:
>
> - **Clarification of "ST-ICL"**: We appreciate your feedback regarding the terminology and agree that it is important to maintain clarity around the definition of ICL as update-less learning. While we intentionally framed our method to emphasize its connection to ICL through the teacher model, we recognize that the current terminology could lead to potential confusion. To address this, we will revise the expanded name of the method to **Self-Training on ICL**, which we believe more accurately reflects its relationship to the teacher model and ensures consistency with established definitions. Additionally, we acknowledge the angle toward fine-tuning that you highlighted. Our method, at its core, is indeed a lightweight fine-tuning approach that integrates the benefits of ICL through the teacher. We will address this in the manuscript to provide a better understanding of the method's positioning.
>
> - **Perspective on disentanglement**: While we understand your suggestion to de-emphasize the perspective on disentanglement, we respectfully maintain our position as outlined in our initial response. We believe that disentangling latent shifts is central to both the practical benefits and the conceptual insights offered by our approach.

---

### Official Review · Reviewer_dJby · 2024-11-12

**Soundness:** 3
**Presentation:** 3
**Contribution:** 3
**Rating:** 6
**Confidence:** 3

**Summary:**

This paper introduces Self-Training ICL (STICL), a novel approach that stabilizes in-context learning (ICL) through self-training. Specifically, STICL uses a student model (LoRA adapter) trained to mimic the output of prompted teacher models, achieving a disentanglement of query and few-shot examples that goes beyond the approximations of previous studies. The authors demonstrate the superior performance and out-of-distribution (OOD) generalizability of this adapter, highlighting its effectiveness over prior ICL methodologies.

**Strengths:**

- Motivation and Clarity: The motivation behind this work is well-explained, with clear distinctions from previous studies. The paper is well-organized and easy to follow, making it accessible to readers.

- Empirical Validation: The paper provides strong empirical evidence supporting the effectiveness of STICL, particularly in OOD settings. This solidifies its contributions and highlights its practical relevance.

**Weaknesses:**

- While Table 3 discusses OOD generalizability, it would be beneficial to include a cross-tabulation across datasets to show the performance in-domain. Understanding the performance gap between in-domain and OOD settings is a key factor that could add depth to the findings.

- Comparison with MetaICL: This study’s use of adapters for STICL could be interpreted as an adaptation of MetaICL, making it important to explicitly clarify the differences between the two approaches. Unfortunately, this work does not appear to cite MetaICL, which may cause readers to miss relevant context and comparisons.

**Questions:**

1. How were the subsets in Table 4 selected? Additional context on this decision would clarify the setup.

2. How are the LoRA parameters configured? Specifically, what level of low-rank approximation is achievable in this setting?

3. Could you clarify the connection between the claim in line 219 and prior studies? A clearer explanation of why this design is impactful would be helpful.

4. In Section 4, the discussion on the Lipschitz constant and weak-to-strong generalization feels somewhat unclear. Could you elaborate on how this contributes to the proposed approach?

---

> ### Author Response · Authors · 2024-11-20
>
> Thank you for your valuable feedback. We are glad that you find the paper well-explained and easy to follow while offering strong empirical evidence supporting the effectiveness of STICL.
>
> ---
>
>  > While Table 3 discusses OOD generalizability, it would be beneficial to include a cross-tabulation across datasets to show the performance in-domain.
>
> We initially chose not to repeat the numbers from Table 1 in Table 3 to maintain conciseness and avoid redundancy. However, we understand that presenting OOD results alongside ID results may aid in comparing the performances directly. We will update the table in the revised manuscript.
>
> ---
>
> > Comparison with MetaICL: This study’s use of adapters for STICL could be interpreted as an adaptation of MetaICL, making it important to explicitly clarify the differences between the two approaches.
>
> We acknowledge that MetaICL is related to our work. We aim to clarify the differences between MetaICL and STICL and explain why we initially chose not to emphasize MetaICL, though we will now include it in the revised manuscript. While both methods aim to enhance ICL through some form of fine-tuning beforehand, they differ significantly in their underlying principles, making MetaICL less central to the focus of our paper. Specifically, we view STICL as more closely related to the disentanglement approach proposed by Dai et al. (2023), which we discuss in the paper. The key distinctions between MetaICL and STICL are:
>
> -   **MetaICL**  fine-tunes the entire model using supervised training across multiple tasks, suited for smaller models like GPT-2 Large (774M). It relies on labeled data for task generalization but does not address latent shifts between demonstrations and queries;
>
> -   **STICL:**  uses a teacher-student self-training framework, generating pseudo-labels for adaptation without additional labeled data. It updates only adapter modules, making it efficient for larger models, and explicitly disentangles latent shifts.
>
> We have now conducted additional experiments with MetaICL and made necessary adaptations to enable a fair comparison with STICL-S. Since STICL uses 100 unlabeled instances, we incorporated their corresponding labels into MetaICL's supervised fine-tuning process to align the setups.
>
> The experiments were conducted on Llama 3 (8B), with configurations of **16 labeled + 100 unlabeled instances** for STICL-S and **116 labeled instances** for MetaICL. We use batches of 16 instances for MetaICL in individual prompts, requiring 8 iterations to complete the fine-tuning for 116 instances. The results (averaged over 10 runs) are as follows:
>
> | Method         | RTE   | SST   | QNLI  | MNLI  | COLA  | MRPC  | QQP   | MATH  | MISC  |
> |----------------|-------|-------|-------|-------|-------|-------|-------|-------|-------|
> | MetaICL        | 82.1  | 95.3  | 79.7  | 71.9  | 62.1  | 75.4  | 72.6  | 45.0  | 84.5  |
> | STICL-S        | 86.0  | 96.1  | 81.4  | 73.1  | 64.3  | 77.7  | 73.1  | 49.5  | 88.0  |
>
> STICL-S outperforms MetaICL across all datasets despite using fewer labeled instances during training. This improvement is likely due to **weak-to-strong generalization** taking effect in STICL-S, where the additional labeled data seems to have a marginal effect on MetaICL. Overall, these results further underscore the potential of combining labeled and unlabeled data through self-training, as demonstrated by STICL. We will include the additional findings in the revised manuscript.
>
> ---
>
> ### Questions
>
> > Q1
>
> We randomly draw subsets of demonstrations from the training set.
>
> ---
>
> > Q2
>
> We use rank r=8 for LoRA. Other configuration parameters are provided in Appendix D.
>
> ---
>
> > Q3
>
> To our knowledge, no prior studies have proposed this exact approach of shuffling demonstrations, likely due to the limited number of works employing fine-tuning to improve ICL. Our motivation arises from well-known challenges in ICL related to demonstration ordering. Liu et al. (2024) observed how LLMs tend to overemphasize information at the beginning and end of a context due to primacy and recency biases. Shuffling demonstrations across epochs can mitigate this issue by varying their positions and stabilizing the ICL process.
>
> ---
>
> > Q4
>
> The relationship between the Lipschitz constant and weak-to-strong generalization may not be immediately clear, so we appreciate the opportunity to clarify. As explained in Section 4.1, a key factor for weak-to-strong generalization is robustness to input variations — the model's ability to maintain stable outputs under small perturbations (Wei et al., 2021; Lang et al., 2024). To test this, we approximated the Lipschitz constant, which bounds the maximum output change for any input change (Khromov & Singh, 2024), by calculating the norms of input-output Jacobian matrices for STICL, PBFT, and ICL. Our results show that STICL has a lower approximated Lipschitz constant, indicating greater stability and providing empirical evidence of its local consistency.

---

> > ### Comment · Reviewer_dJby · 2024-11-20
> >
> > Thank you for the comment. I think the rebuttal about comparison with MetaICL addresses my main concerns and therefore raises the score from 5 to 6.

---

> > > ### Author Response · Authors · 2024-11-20
> > >
> > > Thank you!

---

> > > ### Author Response · Authors · 2024-11-22
> > > **Follow-up**
> > >
> > > We wanted to follow up to ensure that we've fully addressed all of your concerns. If there are any remaining issues or points that require further clarification, we would be happy to provide additional details or adjustments.

---

### Author Response · Authors · 2024-11-27
**Revised manuscript**

We sincerely thank the reviewers once again for their valuable feedback. In response, we have revised and re-uploaded the manuscript with the following additions:
- **Table 3**: Updated to include out-of-distribution (OOD) generalization scores along with their differences from in-distribution (ID) scores.
- **Appendix C**: Expanded to include a detailed discussion of the study's limitations.
- **Appendix D.2**: Added a comparison of STICL with MetaICL.
- **Appendix D.3**: Included an evaluation of STICL in a few-shot setup.
- **Appendix D.4**: Added an analysis of how effectively the student model reconstructs the original demonstrations.

---

### Meta-Review · Area_Chair_jtJD · 2024-12-23

**Metareview:**

This paper introduces a self-training approach purportedly for in-context learning. While all the reviewers initially recommended acceptance, after careful review and thorough discussion, one reviewer recommended rejection. After a comprehensive review, I also agree with the rejection recommendation for the following reasons.

First, the paper writing needs significant improvement. Important technical details are missing in many places, hindering the correct understanding of the proposed methods and claims. The method is also essentially a fine-tuning strategy, which contradicts the framing around in-context learning—the method should be more considered as a fine-tuning method, and one should carefully compare with all the other fine-tuning methods. Furthermore, baseline comparisons appear possibly unfair—given that the method seems to leverage more unlabeled samples than standard ICL, one should also have considered label-free ICL techniques such as zero-shot ICL. The lack of clarity, misleading framing (not ICL), and missing/unfair baselines lead me to recommend rejection. That said, the idea seems promising, but I think the authors may want to resubmit it after correctly positioning it and comparing with proper existing methods.

**Additional Comments On Reviewer Discussion:**

Please see the Metareview.

---

### Decision · Program_Chairs · 2025-01-22

Reject